# DeepThinkVLA: Enhancing Reasoning Capability of Vision-Language-Action Models

## Abstract

Enabling Vision-Language-Action (VLA) models to "think before acting" via Chain-of-Thought (CoT) is a promising path to overcoming the data-hungry nature of end-to-end robot policies. However, progress is stalled by a fundamental conflict: existing models use a single autoregressive decoder for both sequential CoT reasoning and high-dimensional, parallelizable robot actions. This architectural mismatch degrades motor control and fails to forge a strong causal link between thought and action. We introduce DeepThinkVLA, which resolves this conflict through a tightly integrated architecture and training strategy. Architecturally, our hybrid-attention decoder generates sequential CoT with causal attention and then switches to bidirectional attention for fast, parallel decoding of action vectors. This design is complemented by a two-stage training pipeline: we first use Supervised Fine-Tuning (SFT) to teach the model foundational reasoning, then apply Reinforcement Learning (RL) with task-success rewards to causally align the full reasoning-action sequence with desired outcomes. This synergy leads to state-of-the-art performance, achieving a 97.0% success rate on the LIBERO benchmark and demonstrating robust generalization on the high-fidelity RoboTwin 2.0 benchmark, outperforming the strongest baseline by 21.4%.. Our ablations confirm the design's effectiveness: the hybrid architecture alone outperforms standard decoders by 15.5%, and the final RL stage provides a crucial 2% boost to secure top performance. [1]

## 1 Introduction

Vision–Language–Action (VLA) models have driven notable progress in robotic manipulation, enabling tasks like stacking blocks, opening drawers, and arranging household objects (Huang et al., 2023; Zitkovich et al., 2023; Yang et al., 2024; Cadene et al., 2024). The dominant paradigm learns a reactive, end-to-end policy that directly maps high-level goals and sensory inputs to low-level motor commands (Chi et al., 2023; Kim et al., 2024; Bjorck et al., 2025). However, this approach faces a critical bottleneck: learning this monolithic, perception-to-action mapping is inherently difficult and demands vast amounts of high-quality demonstration data (Ma et al., 2024; Liu et al., 2025b).

To address the challenges of direct perception-to-action mapping, one promising direction is to endow VLAs with the ability to "think before acting" (Zawalski et al., 2024; Chen et al., 2025b). Recent studies take initial steps in this direction by supervised fine-tuning (SFT) VLAs on embodied Chain-of-Thought (CoT) datasets (Zawalski et al., 2024). This approach forces the policy to generate CoT reasoning before actions, aiming to simplify the direct mapping problem.

However, a critical limitation of these methods lies in their underlying architecture (Liu et al., 2025d; Sapkota et al., 2025). First, these studies typically employ a unified, purely autoregressive decoder to generate the entire reasoning and action sequence. This design creates a fundamental conflict: CoT, as natural language, is inherently sequential and well-suited for autoregressive modeling (Xiao et al., 2023), whereas actions are often high-dimensional vectors where dimensions (e.g., end-effector translation and rotation) can be determined in parallel and are highly sensitive to latency (Liu et al., 2025d; Kim et al., 2025b; Song et al., 2025). Forcing a single model to master both modalities can compromise motor control precision for the sake of language fluency. While Zhao et al. (2025b)

---

[1]Code available at `https://anonymous.4open.science/r/DeepThinkVLA`

apply parallel decoding to visual subgoals, their approach prioritizes state reconstruction ("what it looks like") over causal reasoning ("why to act"). This reliance limits logical error recovery and hinders the efficacy of RL optimization, which benefits significantly from discrete tokens. Second, relying solely on SFT often leads to the model merely "rote learning" the CoT annotations without establishing a strong causal link to subsequent actions. This results in limited improvement in task success, as the reasoning is not effectively utilized to guide behavior.

To this end, we propose DeepThinkVLA, an approach that synergistically co-designs the model architecture and the training strategy. From the architectural perspective, we design a hybrid-attention decoder. This decoder employs causal attention for generating the sequential COT and switches to bidirectional attention for parallel decoding of the high-dimensional action vector. This design not only respects the intrinsic properties of language and motor commands but also critically reduces the inference latency associated with action generation. From the training perspective, we introduce a RL training pipeline (Zhang et al., 2024; Kim et al., 2025a; Li et al., 2025): first, we perform SFT on an embodied CoT dataset to instill the basic "thinking" ability in the model. Subsequently, leveraging the high-speed rollouts enabled by our parallel action decoder, we apply outcome-based RL. This RL stage directly aligns the entire reasoning-action sequence with task success, optimizing the CoT to be not just plausible, but genuinely beneficial for solving the task.

Based on the public weights of $\pi_0$-FAST (Pertsch et al., 2025), we build DeepThinkVLA by integrating a hybrid architecture, an SFT cold-start on an embodied CoT dataset from our two-stage pipeline, and outcome-based RL. Our experiments provide clear evidence for the effectiveness of the proposed approach. After SFT, applying RL to align CoT reasoning with the generated actions improves task success rates by an additional 2% on the LIBERO-Long suite. The hybrid architecture, designed to support the "think before acting" paradigm, proves critical: training without this modification on the embodied CoT dataset results in an average success rate 15.5% lower than with the hybrid design. Overall, DeepThinkVLA achieves an average success rate of 97.0% on the LIBERO benchmark under the single-model, multi-suite evaluation protocol and further demonstrates robust generalization on the high-fidelity RoboTwin 2.0 benchmark, outperforming the strongest baseline by 21.4%.

## 2 RELATED WORK

**Vision–Language–Action Models.** Recent progress in VLAs has focused on architectural innovations built on the reactive perception-to-action paradigm. Early work such as RT-2 (Zitkovich et al., 2023) popularized this direction, leading to a wide range of models that adopt different VLM backbones (Black et al., 2025; Bjorck et al., 2025; Hung et al., 2025; Pertsch et al., 2025) and are trained on large-scale robotic datasets (Walke et al., 2023; Fang et al., 2024; O'Neill et al., 2024; Khazatsky et al., 2024; Wu et al., 2025). Variants include diffusion-based decoders that model complex action distributions through iterative refinement (Black et al., 2025; Liu et al., 2025a), block-parallel decoders that improve distributional modeling by applying bidirectional attention to predict multiple action tokens (Liu et al., 2025d; Kim et al., 2025b; Song et al., 2025), and hierarchical structures that separate planning from execution (Belkhale et al., 2024; Cui et al., 2025; Team et al., 2025; Bu et al., 2025a; Intelligence et al., 2025). Yet across these designs, the action-generating policy remains reactive—mapping observations and goals directly to actions (or observations and subgoals in hierarchical variants). Consequently, the overall philosophy is still direct mapping, which becomes inadequate under a "think before acting" paradigm that demands both CoT-friendly text generation and high-performance action decoding.

**Embodied Reasoning via Supervised Fine-Tuning.** To move beyond purely reactive mapping, several recent studies have attempted to endow VLAs with embodied reasoning ability through SFT on CoT augmented data (Zawalski et al., 2024; Lin et al., 2025; Tan et al., 2025; Chen et al., 2025b). This line of work typically constructs CoT annotations for existing embodied datasets (O'Neill et al., 2024), often by leveraging stronger cloud-based models (Team, 2023), and then fine-tunes open-source VLAs such as OpenVLA (Kim et al., 2024). While such methods provide a first step toward "think before acting," they face two persistent challenges. First, the availability of high-quality, CoT-annotated embodied data remains limited (Wang et al., 2024; Xu et al., 2024; Zhong et al., 2025). Second, SFT tends to treat reasoning and action generation as parallel objectives within a fixed dataset (Zhao et al., 2025a; Liu et al., 2025c), which can lead to shallow memorization of reasoning

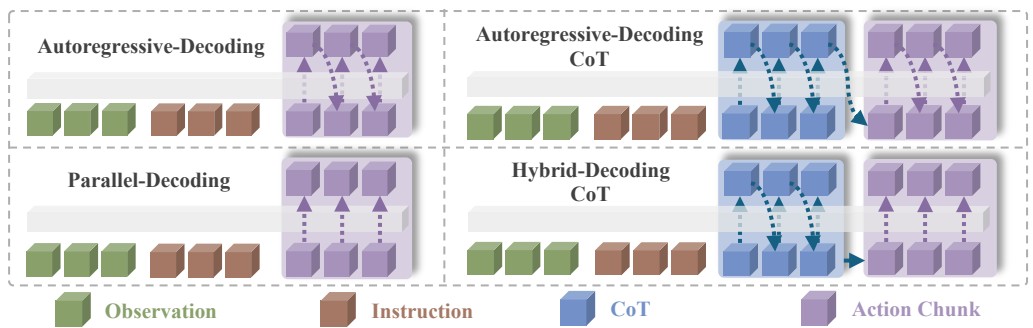

Figure 1: Comparison of VLA architectures. Existing designs adopt either fully autoregressive decoding or parallel bidirectional decoding. DeepThinkVLA introduces a hybrid architecture, enabling autoregressive CoT reasoning alongside efficient parallel action generation.

traces without strong alignment to the resulting actions. As a result, performance improvements from SFT alone are often modest, indicating the need for approaches that can more directly couple reasoning with successful task execution.

## 3 DEEPTHINKVLA

DeepThinkVLA is built on the synergistic co-design of a novel model architecture and an RL training pipeline. We first formalize the "think before acting" paradigm as a probabilistic decomposition (Section 3.1). We then introduce our core architectural innovation, the hybrid-attention decoder, which is specifically designed to implement this decomposition efficiently (Section 3.2). Finally, we detail our RL training strategy, where this architecture first enables foundational reasoning via SFT and then makes large-scale RL practical for aligning this reasoning with task success (Section 3.3).

### 3.1 PROBLEM FORMULATION

Standard VLA policies learn a direct mapping from visual observations ($V$) and language instructions ($L$) to a sequence of actions ($A$). Instead, our work adopts the principle of "think before acting". We implement this principle by introducing a latent reasoning variable, the Chain-of-Thought ($R$). This approach decomposes the problem. Rather than modeling the direct policy, we model the joint probability of reasoning and then acting:

$$P(A, R|V, L) = P(A|V, L, R)\, P(R|V, L). \tag{1}$$

The advantage of this decomposition is twofold. First, learning $P(R|V, L)$ is highly efficient. Most VLAs are built upon large VLM backbones, which already contain rich semantic and reasoning knowledge. Fine-tuning such models on a relatively small set of synthetically generated embodied CoT data is often sufficient to adapt their reasoning capability to the robotics domain. Second, learning $P(A|V, L, R)$ becomes significantly simpler compared to directly modeling $P(A|V, L)$. The CoT $R$ serves as an explicit, step-by-step plan that disambiguates high-level instructions $L$, turning the original ill-posed, one-to-many mapping into a constrained and well-specified mapping from a reasoning step to its corresponding motor action. This principled factorization also enables emergent self-correction behaviors, as illustrated in Figure 6.

### 3.2 HYBRID ARCHITECTURE

To implement the factorized policy from Eq. 1, we introduce a novel hybrid architecture. Its primary motivation is to resolve the fundamental conflict between the modalities of reasoning and action within a unified autoregressive decoder.

We propose a novel hybrid architecture that aligns the decoding mechanism with the intrinsic properties of each modality. The core of our design is a dynamic attention mode within a single decoder:

For CoT Generation ($P(R|V, L)$): The decoder employs standard autoregressive causal attention. This respects the sequential nature of language, where each reasoning token is generated based on its predecessors. For Action Generation ($P(A|V, L, R)$): After generating the CoT, the attention mechanism switches to bidirectional (non-causal) attention. This allows the model to process the entire action specification jointly and decode the action vector in parallel, acknowledging that different dimensions of a motor command (e.g., translation and rotation) are often determined concurrently.

Beyond resolving this core modality mismatch, the parallel decoding of actions yields a critical practical advantage: a significant reduction in inference latency. This speedup is the key enabler for our subsequent training stage. While standard autoregressive models are prohibitively slow for the massive number of rollouts required by on-policy Reinforcement Learning, our architecture's high-throughput action generation makes large-scale online fine-tuning computationally tractable. Further architectural details are summarized in Figure 1.

### 3.3 RL TRAINING PIPELINE

With our efficient architecture in place, we train DeepThinkVLA using a two-stage pipeline designed to first instill foundational reasoning and then align it with optimal task outcomes.

**SFT Cold-Start for Foundational Reasoning** SFT cold-start is designed to equip the model with a foundational CoT reasoning capability. This phase requires a specific supervision format comprising complete $(V, L, R, A)$ sequences. However, most existing large-scale embodied datasets lack explicit CoT annotations and instead provide only $(V, L, A)$ tuples. To address this critical data gap, we developed a scalable, two-stage data augmentation pipeline that generates high-quality CoT annotations, as illustrated in Figure 2.

Our pipeline is optimized for both annotation quality and cost-efficiency. In stage 1, we identify semantically significant keyframes within each trajectory by detecting changes in the gripper state, which often indicate subtask boundaries. For these keyframes, CoT annotations are obtained by querying a powerful, general-purpose cloud-based Vision-Language Model (VLM) (see prompt in Appendix Fig. 5). In stage 2, to efficiently annotate the numerous intermediate frames, we fine-tune a smaller, locally-deployed VLM on the high-quality keyframe annotations obtained in Stage 1. This specialized model then automatically generates CoT annotations for the transitional frames. To ensure data fidelity, we apply schema checks to filter malformed outputs and enforce temporal consistency, resulting in a uniform embodied CoT dataset suitable for SFT.

**Learning Reasoning and Action via RL** While SFT provides a strong imitation learning foundation, it cannot adapt to novel scenarios or optimize beyond the sub-optimal trajectories present in the static dataset. To overcome these limitations and truly learn a policy that maximizes task success, we introduce a second training stage based on online Reinforcement Learning. We use an outcome-based reward to jointly optimize the entire reasoning-action sequence towards the singular goal of task completion (Figure 3).

We cast the online RL stage as a policy optimization problem: starting from the initial SFT policy, we aim to maximize the expected outcome-based reward through interactive learning. We adopt an on-policy policy gradient algorithm with a clipped surrogate objective akin to PPO. The VLA collects trajectories from the environment, computes advantages, and updates its policy accordingly.

Formally, during the RL rollout, the state at each step $t$ is defined as $s_t = [o_t^{\text{vis}}, \ell_{\text{task}}]$, where $o_t^{\text{vis}}$ denotes the visual observation and $\ell_{\text{task}}$ is the task instruction. Given the state input, the VLA outputs $\mathcal{A}_t = [a_t^{\text{cot}}, a_t^{\text{robot}}]$, where the reasoning tokens $a_t^{\text{cot}}$ are generated autoregressively, and the action tokens $a_t^{\text{robot}} \in \mathbb{R}^{h \times d}$ are decoded in parallel. Here, $h$ denotes the action chunk size and $d$ corresponds to the robot's control dimension (e.g., $d = 7$ for a 6-DoF manipulator plus gripper control).

Let $\pi_\theta$ denotes the current policy and a trajectory $\tau$ sampled from the old policy $\pi_{\theta_{\text{old}}}$ is defined as $\tau = [(s_0, \mathcal{A}_0), (s_1, \mathcal{A}_1), \ldots, (s_T, \mathcal{A}_T)]$. The reward function $\mathcal{R}(\tau)$ is sparse, awarded only at the end of the trajectory based on a verifiable task completion signal $\mathcal{I}_{\text{success}}$. No intermediate reward is given for the semantics of the reasoning trace. A small format-regularization reward $\mathcal{I}_{\text{format}}$ is added

Figure 2: Pipeline for constructing an embodied CoT dataset. Stage 1 extracts keyframes via gripper state signals and queries a cloud LVLM to generate CoT for those frames. Stage 2 fine-tunes a local vision–language model on the keyframe CoT and uses it to annotate the remaining frames.

to prevent stylistic drift. Hence, the reward function is defined as

$$\mathcal{R}(\tau) = \alpha_s \cdot \mathcal{I}_{\text{success}} + \alpha_f \cdot \mathcal{I}_{\text{format}},$$

$$\mathcal{I}_{\text{success}} = \begin{cases} 1, & \text{if task success,} \\ 0, & \text{otherwise,} \end{cases} \qquad \mathcal{I}_{\text{format}} = \begin{cases} 1, & \text{if CoT format correct,} \\ 0, & \text{otherwise.} \end{cases} \tag{2}$$

where $\alpha_s$ and $\alpha_f$ are weighting coefficients. Then the token-level clipped surrogate objective is:

$$\mathcal{J}(\theta) = \mathbb{E}_{\tau \sim \pi_{\theta_{\text{old}}}} \left[ \sum_{j=1}^{N} \min\Big( \omega_j(\theta)\, \hat{A}_j,\ \text{clip}\big(\omega_j(\theta), 1-\epsilon, 1+\epsilon\big)\, \hat{A}_j \Big) \right], \tag{3}$$

where $N = |\mathcal{A}_t| \times T$ denotes the total number of tokens in a trajectory $\tau$. $\omega_j(\theta) = \frac{\pi_\theta(a_j | s_t, a_{<j})}{\pi_{\theta_{\text{old}}}(a_j | s_t, a_{<j})}$ and $\hat{A}_j$ are the importance ratio and the advantage for token $a_j \in [\mathcal{A}_0, \ldots, \mathcal{A}_T]$ within trajectory $\tau$, respectively.

To propagate the sparse, outcome-based reward $\mathcal{R}(\tau)$ to each token prediction, we adopt the credit assignment strategy from GRPO (Shao et al., 2024). A group of $G$ trajectories is collected for each task prompt and their rewards are standardized to compute a shared advantage value for all tokens within a given trajectory. For token $j$ in trajectory $i$, the advantage is:

$$\hat{A}_{i,j} = \frac{\mathcal{R}(\tau_i) - \text{mean}\big(\{\mathcal{R}(\tau_k)\}_{k=1}^{G}\big)}{\text{std}\big(\{\mathcal{R}(\tau_k)\}_{k=1}^{G}\big)}. \tag{4}$$

This relative credit assignment encourages the model to prefer reasoning and action sequences that lead to better-than-average outcomes, effectively selecting for more functional thought processes.

Combining the clipped surrogate objective with the GRPO-style advantage and a KL-divergence penalty to the original SFT policy $\pi_{\text{ref}}$, which prevents catastrophic forgetting, our final objective is:

$$\mathcal{J}_{\text{final}}(\theta) = \mathbb{E}_{s \sim \text{env},\, \{\tau_i\}_{i=1}^{G} \sim \pi_{\theta_{\text{old}}}} \left[ \frac{1}{G} \sum_{i=1}^{G} \frac{1}{N} \sum_{j=1}^{N} \min\Big( \omega_{i,j}(\theta)\, \hat{A}_{i,j}, \right.$$

$$\left. \text{clip}\big(\omega_{i,j}(\theta), 1-\epsilon, 1+\epsilon\big)\, \hat{A}_{i,j} \Big) - \beta\, \text{KL}\big(\pi_\theta(\cdot \mid s) \,\|\, \pi_{\text{ref}}(\cdot \mid s)\big) \right], \tag{5}$$

Figure 3: Reinforcement learning stage with grouped credit assignment. The model generates CoT and action sequences that are executed in the simulator to produce trajectories with verifiable rewards. Rewards are grouped and standardized to compute token-level advantages, which update the policy via a clipped surrogate objective with KL regularization to the SFT reference.

Table 1: **Main results on the LIBERO Simulation Benchmark.** All reported values denote the task success rate (SR, %) evaluated under 50 randomized initial conditions per task, averaged within each suite and across all suites. All top-performing baselines utilize wrist camera inputs to ensure fair comparison; notably, DeepThinkVLA achieves the reported results without proprioception. **AR** denoted as Autoregressive. **Bold numbers** indicate the best performance within each suite.

| Category | Method | Object | Spatial | Goal | Long | Average |
|---|---|---|---|---|---|---|
| **AR-Decoding** | TraceVLA (Zheng et al., 2024) | 85.2 | 84.6 | 75.1 | 54.1 | 74.8 |
| | Octo (Team et al., 2024) | 85.7 | 78.9 | 84.6 | 51.1 | 75.1 |
| | OpenVLA (Kim et al., 2024) | 88.4 | 84.7 | 79.2 | 53.7 | 76.5 |
| | SpatialVLA (Qu et al., 2025) | 89.9 | 88.2 | 78.6 | 55.5 | 78.1 |
| | GRAPE (Zhang et al., 2024) | 91.2 | 87.6 | 82.2 | 55.8 | 79.2 |
| | NORA (Hung et al., 2025) | 95.4 | 92.2 | 89.4 | 74.6 | 87.9 |
| | VLA-RL (Lu et al., 2025) | 91.8 | 90.2 | 82.2 | 59.8 | 81.0 |
| | $\pi_0$-FAST (Pertsch et al., 2025) | 96.8 | 96.4 | 88.6 | 60.2 | 85.5 |
| | UniVLA (Bu et al., 2025b) | 96.8 | 96.5 | 95.6 | 92.0 | 95.2 |
| **Diffusion** | Diffusion Policy (Chi et al., 2023) | 92.5 | 78.3 | 68.3 | 50.5 | 72.4 |
| | $\pi_0$ (Black et al., 2025) | 98.8 | **96.8** | 95.8 | 85.2 | 94.2 |
| **Parallel-Decoding** | CoT-VLA-7B (Zhao et al., 2025b) | 91.6 | 87.5 | 87.6 | 69.0 | 81.1 |
| | OpenVLA–OFT (Kim et al., 2025b) | 92.7 | 91.3 | 90.5 | 86.5 | 90.3 |
| **Hybrid-Decoding** | DeepThinkVLA (Ours) | **99.0** | 96.6 | **96.4** | **96.2** | **97.0** |

where $\omega_{i,j}(\theta)$ is the the importance ratio for token $a_j \in [\mathcal{A}_0, \ldots, \mathcal{A}_T]$ in trajectory $i$ By maximizing $\mathcal{J}_{\text{final}}$, the VLA simultaneously refines its reasoning and action abilities, with both aligned toward the singular goal of maximizing the final task success rate.

## 4 EXPERIMENTS

### 4.1 EXPERIMENTAL SETUP

**Implementation Details.** DeepThinkVLA is initialized from the public $\pi_0$-FAST weights (Pertsch et al., 2025). We refactor the baseline policy with our hybrid-attention decoder (Sec. 3.2), yielding a 2.9B parameter model. Training proceeds in two stages. First, an SFT cold-start is performed on our embodied CoT dataset (Sec. 3.3). A hybrid attention mask supervises CoT tokens with causal attention and action tokens with bidirectional attention in a single forward pass, optimized by token-level cross-entropy loss. Second, we apply online RL (Sec. 3.3) to align generated CoT with action execution, using outcome-based rewards and GRPO-style grouped credit

Table 2: **Comparison of various VLA models on the high-fidelity RoboTwin 2.0 benchmark.** DeepThinkVLA significantly outperforms baselines, particularly on long-horizon tasks.

| Short Horizon Tasks (100–130 Steps) | | | | | |
|---|---|---|---|---|---|
| **Model** | **Lift Pot** | **Beat Hammer Block** | **Pick Dual Bottles** | **Place Phone Stand** | **Avg** |
| $\pi_0$ | 51.0 | 59.0 | 50.0 | 22.0 | 45.5 |
| RDT | 45.0 | 22.0 | 18.0 | 13.0 | 24.5 |
| OpenVLA-OFT | 10.1 | 28.1 | 29.7 | 17.1 | 21.3 |
| $\pi_0$-FAST | 30.0 | 38.0 | 25.0 | 16.0 | 27.3 |
| **DeepThinkVLA** | **62.0** | **73.0** | **61.0** | **24.0** | **55.0** |
| $\Delta$ | +32.0 | +35.0 | +36.0 | +8.0 | +27.8 |
| Medium Horizon Tasks (150–230 Steps) | | | | | |
| **Model** | **Move Can Pot** | **Place A2B Left** | **Place Empty Cup** | **Handover Mic** | **Avg** |
| $\pi_0$ | 41.0 | 38.0 | 60.0 | 96.0 | 58.8 |
| RDT | 33.0 | 21.0 | 42.0 | 95.0 | 47.8 |
| OpenVLA-OFT | 28.1 | 37.5 | 77.3 | 45.3 | 47.1 |
| $\pi_0$-FAST | 34.0 | 36.0 | 54.0 | 83.0 | 51.8 |
| **DeepThinkVLA** | **52.0** | **38.0** | **83.0** | **88.0** | **65.3** |
| $\Delta$ | +18.0 | +2.0 | +29.0 | +5.0 | +13.5 |
| Long (280–320 Steps) & Extra Long Horizon Tasks (450–650 Steps) | | | | | |
| **Model** | **Handover Block** | **Stack Bowls Two** | **Blocks Rank Rgb** | **Put Bottles Dustbin** | **Avg** |
| $\pi_0$ | 39.0 | 53.0 | 45.0 | 36.0 | 43.3 |
| RDT | 26.0 | 42.0 | 17.0 | 26.0 | 27.8 |
| OpenVLA-OFT | 33.1 | 40.6 | 70.2 | 42.2 | 46.5 |
| $\pi_0$-FAST | 32.0 | 48.0 | 28.0 | 27.0 | 33.8 |
| **DeepThinkVLA** | **43.0** | **62.0** | **77.0** | **49.0** | **57.8** |
| $\Delta$ | +11.0 | +14.0 | +49.0 | +22.0 | +24.0 |
| **Overall Avg** RDT: 33.3 $\pi_0$: 49.2 OpenVLA-OFT: 38.3 $\pi_0$-FAST: 38.0 **DeepThinkVLA: 59.4** +21.4 | | | | | |

assignment. Additional training details, including dataset statistics, hyperparameters, and inference settings, are provided in Appendix A.2.

**Benchmarks.** We evaluate DeepThinkVLA on two distinct benchmarks to ensure a comprehensive assessment of both foundational skills and long-horizon robustness. LIBERO (Liu et al., 2023): A standard language-conditioned manipulation benchmark containing four suites (Object, Spatial, Goal, and Long). Evaluation is conducted under 50 randomized initial conditions for each task. RoboTwin 2.0 (Chen et al., 2025a): A high-fidelity digital twin benchmark featuring complex contact-rich manipulation and significantly longer task horizons. This is introduced to rigorously test the model's capability to maintain reasoning context over extended execution periods. Performance is measured by the average success rate. Additionally, to provide concrete examples of the model's reasoning process and error recovery behaviors (Case Studies), we include a detailed qualitative analysis in Appendix A.5.

**Baselines.** We evaluate DeepThinkVLA against a wide spectrum of recent VLA systems, covering autoregressive SFT models, diffusion-based approaches, parallel-decoding methods and commercial baselines such as $\pi_0$ and $\pi_0$-FAST. A detailed description of all baselines is provided in Appendix A.3.

## 4.2 MAIN RESULTS

**Results on LIBERO.** The main results on the LIBERO benchmark are shown in Table 1. DeepThinkVLA establishes a new state-of-the-art by achieving the highest average success rate (SR) of 97.0%. Our model shows exceptional proficiency across all categories, particularly in Object (99.0%) and long-horizon tasks (96.2%), significantly outperforming leading diffusion models like $\pi_0$ (94.2%) and autoregressive baselines like UniVLA (96.2%). It is important to note that this com-

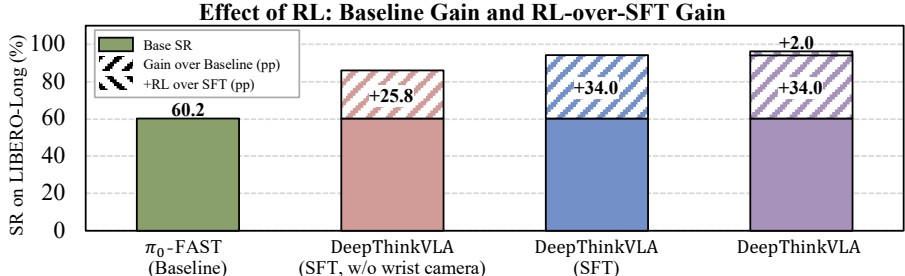

Figure 4: Effect of RL on long-horizon task performance (LIBERO-Long). Bars show base SR for each model, while lighter shaded segments indicate gains over the baseline. For DeepThinkVLA, the additional teal segment highlights the extra improvement from RL over SFT (+2 pp). The figure illustrates that all DeepThinkVLA variants outperform the baseline, and RL further aligns CoT reasoning with action generation to boost success rate.

parison is conducted under strict fairness: all competitive baselines in Table 1 utilize wrist camera inputs. Furthermore, unlike many baselines that additionally rely on proprioceptive state (e.g., joint angles), DeepThinkVLA achieves this performance using a pure vision-language setup. The substantial gap over other parallel-decoding methods (e.g., CoT-VLA-7B at 81.1%) further underscores that our specific Hybrid-Decoding design is critical for unlocking high-performance reasoning.

**Results on RoboTwin 2.0.** To further evaluate the model's robustness on tasks with high-fidelity physics and significantly longer horizons, we extended our evaluation to the RoboTwin 2.0 benchmark (Table 2). DeepThinkVLA demonstrates strong generalization capabilities, achieving an average success rate of 59.4%, which significantly outperforms the strong $\pi_0$-FAST baseline (38.0%) and the generic $\pi_0$ model (49.2%). Most notably, on the challenging "Extra Long Horizon" tasks which require maintaining context over hundreds of steps, our method maintains robust performance (57.8%) where purely reactive baselines falter (33.8%). This confirms that DeepThinkVLA does not merely overfit to the LIBERO distribution but possesses the generalized "Think-Before-Acting" capability required for complex, dynamic environments.

## 4.3 ANALYSIS EXPERIMENT

**RL for CoT–Action Alignment.** We first examine the impact of our outcome-based Reinforcement Learning stage. As shown in Figure 4, applying RL increases the success rate on the challenging LIBERO-Long suite from 94.2% (SFT-only) to 96.2%. While this +2.0% gain validates the effectiveness of alignment, the improvement is constrained by the saturation of the LIBERO benchmark. To rigorously verify the benefits of RL in more complex, contact-rich dynamics, we extended this ablation to the RoboTwin 2.0 benchmark (detailed breakdown provided in Appendix A.6). In this high-fidelity setting, the efficacy of RL becomes significantly more pronounced: DeepThinkVLA(RL) achieves an overall average improvement of +6.8% over the SFT baseline. This contrast confirms that outcome-based optimization is crucial for refining the policy beyond the limitations of static demonstrations, particularly in complex environments.

In a complementary finding regarding sensor modalities, incorporating a wrist camera provides additional improvements by capturing near-field contact information. Crucially, however, even without the wrist camera, our method achieves an 86.0% success rate on LIBERO, which already outperforms the strong $\pi_0$-FAST baseline (85.5%). This result confirms that our performance gains primarily stem from the hybrid architecture and reasoning capability rather than specific sensor modalities.

**Deconstructing Robustness via Performance Drops.** While standard benchmarks demonstrate high success rates, they often fail to distinguish between trajectory memorization and true functional reasoning. To rigorously deconstruct the source of our model's capability, Table 3 presents a comprehensive factorial analysis comparing DeepThinkVLA against the $\pi_0$-FAST baseline across training stages and inference modes under OOD (Joint-Limit) dynamics. The standard $\pi_0$-FAST

Table 3: **Comprehensive Factorial Analysis.** We compare the Baseline against DeepThinkVLA across Training Stages and Inference Modes. The Drop column quantifies the performance loss when shifting from Standard to OOD dynamics.

| Method / Stage | Inference Mode | Dynamics | Success Rate (%) | | | | | Drop |
|---|---|---|---|---|---|---|---|---|
| | | | Obj | Spa | Goal | Long | Avg | (pp) |
| $\pi_0$-FAST Pertsch et al. (2025) | - | Standard | 96.8 | 96.4 | 88.6 | 60.2 | 85.5 | - |
| | | OOD Limit | 77.0 | 64.0 | 31.2 | 43.6 | 53.9 | 31.6 |
| DeepThinkVLA(SFT-Only) | Full CoT | Standard | 99.0 | 97.2 | 96.8 | 94.2 | 96.8 | - |
| | | OOD Limit | 87.0 | 61.4 | 54.2 | 56.4 | 64.8 | 32.0 |
| | Mask CoT | Standard | 99.0 | 97.2 | 95.8 | 93.4 | 96.4 | - |
| | | OOD Limit | 85.2 | 58.0 | 50.6 | 48.2 | 60.5 | 35.9 |
| **DeepThinkVLA(RL-Aligned)** | **Full CoT** | Standard | **99.0** | **96.6** | **96.4** | **96.2** | **97.0** | - |
| | | **OOD Limit** | **91.6** | **64.2** | **66.2** | **68.4** | **72.6** | **24.4** |
| | Mask CoT | Standard | 99.0 | 97.2 | 96.0 | 93.6 | 96.5 | - |
| | | OOD Limit | 88.2 | 60.2 | 61.0 | 65.6 | 68.8 | 27.7 |

baseline achieves respectable performance (85.5%) in standard settings but collapses in the OOD Joint-Limit test (53.9%), suffering a massive 31.6% drop. Crucially, despite achieving high standard performance (96.8%), our DeepThinkVLA(SFT-Only) model mirrors this behavior almost exactly, suffering a 32.0% drop in OOD. This empirical evidence proves that SFT primarily fits the kinematic distribution of the training data. Without RL, the CoT reasoning remains merely descriptive—mimicking the expert's kinematics without learning the functional resilience required to adapt.

The DeepThinkVLA(RL-Aligned) model with Full CoT is the configuration that breaks this fragility pattern, limiting the performance drop to 24.4%. This represents a relative improvement in robustness of over 7 percentage points compared to the baseline and SFT models. By optimizing for task success rather than trajectory likelihood, the RL stage enables the agent to re-plan effectively when execution is constrained. The role of inference-time reasoning is further isolated by comparing Full CoT and Mask CoT within the RL-Aligned stage. When CoT is masked, the drop increases from 24.4% to 27.7%. This 3.3% gap confirms that the robustness is not solely implicitly stored in the weights (System-1) but relies on the explicit generation of reasoning tokens (System-2) to navigate out-of-distribution dynamics. For further analysis on architectural choices (Hybrid vs. AR) and the semantic role of CoT (Mask vs. Random), please refer to Appendix A.4.

## 5    CONCLUSIONS AND FUTURE WORK

This work addresses critical challenges in the "think before acting" paradigm for VLA models. We identify two fundamental limitations in existing methods. First, they face an architectural conflict, using a single autoregressive decoder for both sequential reasoning and parallel motor commands. Second, they suffer from a training deficiency, where the generated reasoning is not causally linked to task success. To resolve these issues, we introduce DeepThinkVLA, an integrated approach that co-designs the model architecture and training strategy. Architecturally, we propose a hybrid-attention decoder that uses causal attention for sequential CoT generation and bidirectional attention for parallel action decoding. For training, we employ a two-stage pipeline: an initial SFT stage instills foundational reasoning, followed by an outcome-based RL stage that aligns this reasoning with task goals. Our empirical results demonstrate the effectiveness of this synergistic design. The hybrid architecture is essential for leveraging CoT and significantly outperforms a naive autoregressive baseline. Furthermore, the RL stage yields additional performance gains by optimizing reasoning for functional utility. Collectively, these innovations enable DeepThinkVLA to establish a new SOTA on the LIBERO and RoboTwin2.0 benchmark. Our work underscores that integrating explicit reasoning into robotics requires a holistic approach that tightly aligns model architecture with training objectives.

## ETHICS STATEMENT

This work uses publicly available, license-compliant datasets and simulated environments only; no personally identifiable information, human subjects, or animal experiments are involved. For embodied settings, all evaluations are conducted in simulation with safety constraints (termination conditions, joint/force limits, and reset policies) to avoid unsafe behaviors; any real-world deployment should include additional risk assessments and hardware interlocks.

We acknowledge that foundation models may carry societal biases. To mitigate this, we employ schema checks and consistency filters during data construction, report failure cases, and avoid tasks with sensitive attributes. Our method could be misused to automate unsafe manipulation; we strongly discourage use in safety-critical contexts without appropriate supervision and compliance.

We report configurations and code to facilitate reproducibility. There are no conflicts of interest or undisclosed sponsorships. The authors have read and will adhere to the Code of Ethics.

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

# A    APPENDIX

## A.1    PROMPT OF AUTOMATED PIPELINE FOR DATA CONSTRUCTION

```
Role Assignment
You are an advanced robotic intelligence agent.
Inputs
●  Global Task Instruction - overall goal.
●  Keyframes - an ordered list of N external-camera images capturing
     critical moments.
- Frame 1 is before any subtask; Frame i (i > 1) is after subtask i
     -1 and before subtask i.
Required Output
For each keyframe (in order), output exactly:
Produce exactly N consecutive (<think> ... </think><subtask> ... </
     subtask>) pairs--one pair per image, same order, plain text, no
     extra lines. (≤ 50 words, single paragraph, no line breaks)
<think> rules
1. Frame 1: declare initial frame.
2. Frame i (i > 1): internally compare the previous image with the
     current image to infer the effect of the last subtask, but **do
     not state Success or Failure explicitly**.
3. Tag every instruction-relevant object with its location in the
     current image, e.g., bowl (right-front).
4. Describe spatial layout, affordances, obstacles, and reasoning
     that motivates an immediate next subgoal advancing the Global
     Task.
5. Do not mention other frames, prior success, failure, or progress
     metrics. Just analyze the current frame.
6. Do not mention frame indices like Frame 1; use initial frame.
<subtask> rules
●  State the immediate next subgoal in clear natural language, **
     without numerals or explicit quantities**.
●  Each subtask must correspond to the change that should occur
     between the current image and the next one (or to completion at
     the final image).
●  If the Global Task is fulfilled in the current image, output
     exactly <subtask>finish</subtask> and stop; omit remaining pairs
     .
Global Constraints
●  Output pairs equal exactly the number of keyframes provided.
●  No extra text outside the mandated tags.
●  No bullet points or lists.
●  No numbers inside any <subtask> tag.
●  Reasoning and subtasks must align with the images, temporal flow,
     and the global instruction.
```

Figure 5: Prompt for Constructing CoT Data at Keyframes Using a Cloud-based LVLM: The prompt is designed to guide the LVLM in generating CoT data at keyframes, primarily focusing on two aspects: (i) describing the spatial relationships present in the visual scene, and (ii) capturing the temporal relationships of subtasks between consecutive keyframes.

## A.2    IMPLEMENTATION DETAILS

DeepThinkVLA is initialized from the public $\pi_0$-FAST weights (Pertsch et al., 2025). We refactor the baseline policy with our hybrid-attention decoder (Sec. 3.1), yielding a 2.9B parameter model.

**Dataset Construction.**    Before SFT, we construct an embodied CoT dataset using the two-stage pipeline described in Sec. 3.3, based on the public LIBERO demonstrations. This process yields 273,465 annotated frames, which serve as the supervision source for the cold-start stage.

**Supervised Fine-Tuning (SFT).**    For the SFT stage, we train with a batch size of 128 and a learning rate of $2.5 \times 10^{-5}$ for 150k steps. A hybrid attention mask is employed so that CoT tokens are supervised autoregressively while action tokens are supervised bidirectionally within the same forward pass. Optimization is performed using a standard token-level cross-entropy loss.

**Reinforcement Learning (RL).**    For the RL stage, we adopt Group Relative Policy Optimization (GRPO) (Shao et al., 2024). The action chunk size is set to 10. Each trajectory receives a sparse task-success reward plus a small format-regularization reward to maintain CoT quality. Policy updates use a mini-batch size of 128, a low clip ratio $\epsilon = 0.2$, a high clip ratio $\epsilon = 0.28$, and a KL penalty to the SFT reference model to avoid catastrophic forgetting.

**Infrastructure and Inference.**    Training is conducted on $8 \times$NVIDIA A800 GPUs. At inference time, we use greedy decoding for both reasoning and action tokens. For CoT ablation experiments, we evaluate three inference modes: Full CoT, Mask CoT, and Random CoT.

### A.3    BASELINES DETAILS

We benchmark DeepThinkVLA against a representative set of VLA systems covering multiple decoding styles, as listed in Table 1.

**Autoregressive SFT models.**    TraceVLA (Zheng et al., 2024) enhances spatial-temporal awareness by overlaying visual trace cues on past motion trajectories to guide action prediction. Octo (Team et al., 2024) is a generalist VLA trained with diverse task distributions under autoregressive decoding. OpenVLA (Kim et al., 2024) is a widely used open-source baseline mapping vision and language directly to actions in autoregressive form. SpatialVLA (Qu et al., 2025) explicitly incorporates spatial reasoning modules into the autoregressive action policy. NORA (Hung et al., 2025) is a compact (3B) VLA model that reduces inference overhead while maintaining task performance. UniVLA (Bu et al., 2025b) seeks a unified generalist agent by training on multiple task suites under the AR decoding paradigm. GRAPE (Zhang et al., 2024) augments AR VLAs with gradient-based policy optimization to refine behavior beyond demonstration data. VLA-RL (Lu et al., 2025) applies outcome-driven RL on top of AR policies to better align actions with task success. $\pi_0$-FAST (Pertsch et al., 2025) is an autoregressive VLA built on the FAST action tokenizer, which enables efficient training of high-frequency dexterous skills while retaining competitive performance.

**Diffusion-based policies.**    Diffusion Policy (Chi et al., 2023) models actions via iterative denoising steps, enabling richer multi-modal action generation. $\pi_0$ (Black et al., 2025) is a flow/diffusion-style VLA that decodes continuous control signals through a diffusion process.

**Parallel/Block-Decoding policies.**    CoT-VLA-7B (Zhao et al., 2025b) employs a block-parallel decoding strategy to generate action tokens in chunks, while still supporting CoT supervision. OpenVLA–OFT (Kim et al., 2025b) uses bidirectional attention over action tokens to accelerate inference and improve accuracy.

**Industrial AR references.**    We also include strong, high-performance autoregressive systems from practice: , and $\pi_0$ (Black et al., 2025) serves as a flow/diffusion alternative reference in real-world settings.

This selection spans autoregressive, diffusion, and parallel-decoding paradigms from both academic and industry sources, under the evaluation of LIBERO and RoboTwin2.0.

### A.4    ABLATIONS ON ARCHITECTURE AND COT ROLE

Table 4: **Ablation on CoT and hybrid-decoding.** All values except the last column denote success rate (%). Methods include the baseline $\pi_0$-FAST, a naive AR-CoT variant, and our hybrid-Decoding DeepThinkVLA. **AR** denoted as Autoregressive. **Bold numbers** indicate the best performance within each suite.

| Category | Method | Object (%) | Spatial (%) | Goal (%) | Long (%) | Average (%) | Rel. Inference Time |
|---|---|---|---|---|---|---|---|
| Baseline | $\pi_0$-FAST | 96.8 | 96.4 | 88.6 | 60.2 | 85.5 | 1.0$\times$ |
| AR-CoT | $\pi_0$-FAST (Full CoT) | 95.8 | 93.8 | 74.6 | 61.0 | 81.3 | 4.0$\times$ |
| Hybrid CoT | DeepThinkVLA (Mask CoT) | 99.0 | 97.2 | 96.0 | 93.6 | 96.5 | 0.175$\times$ |
| | DeepThinkVLA (Random CoT) | 97.8 | 94.4 | 60.2 | 87.8 | 85.1 | 0.175$\times$ |
| | DeepThinkVLA (Full CoT) | **99.0** | **97.2** | **96.8** | **94.2** | **96.8** | 1.4$\times$ |

**Architecture Matters for the Think-Before-Acting Paradigm.** To examine how architectural choices influence the effectiveness of the "think before acting" paradigm, we conducted experiments comparing different model structures under CoT supervision. Specifically, we applied CoT supervision directly to the autoregressive baseline $\pi_0$-FAST and compared its performance with our proposed hybrid architecture.

As shown in Table 4, the autoregressive baseline with CoT supervision ($\pi_0$-FAST (Full CoT)) underperforms the original model (Average: 81.3% vs. 85.5%) and suffers a $4\times$ increase in inference latency. This indicates that forcing a single autoregressive decoder to jointly generate reasoning and actions introduces interference. By contrast, our hybrid architecture achieves markedly better results (Average: 96.8%), representing a +15.5 percentage point gain over the naive AR-CoT baseline with only modest latency overhead.

**The Role of Semantic Coherence.** To further clarify whether CoT serves as an explicit reasoning guidance mechanism, we evaluated the Random CoT condition, where CoT tokens are substituted with randomly generated tokens. As reported in Table 4, while the Mask CoT variant shows almost no performance loss in standard settings (96.5%), the Random CoT setting leads to a sharp decline in success (Average: 85.1%). This indicates that once reasoning traces are utilized during inference, their semantic coherence is essential for successful execution; disrupting this coherence with random noise actively harms the policy.

### A.5 QUALITATIVE CASE STUDY

To illustrate the practical benefits of our approach, we conduct a case study comparing the behavior of DeepThinkVLA with the $\pi_0$-FAST baseline, as depicted in Figure 6. The baseline policy successfully approaches the object but fails to secure a proper grasp. Following this error, it enters a repetitive failure loop, executing unproductive motions without making meaningful progress. This behavior highlights the inherent limitation of direct perception-to-action mapping: once an execution error occurs, the policy lacks an internal mechanism to recover, leading to task failure.

In contrast, DeepThinkVLA generates a concise `<think>...</think>` reasoning trace before each action chunk. When an accidental drop occurs, the model leverages its CoT to restate the subgoal (e.g., "the butter needs to be moved into the basket"), enabling it to reattempt the grasp and successfully complete the task. This demonstrates that our proposed method has effectively aligned reasoning with actions, rendering the CoT not just descriptive but actionable for guiding corrective behaviors. This case study highlights a key qualitative difference: while purely reactive policies are brittle to perturbations, DeepThinkVLA leverages CoT reasoning as an explicit planning and recovery scaffold, enabling more stable and resilient execution.

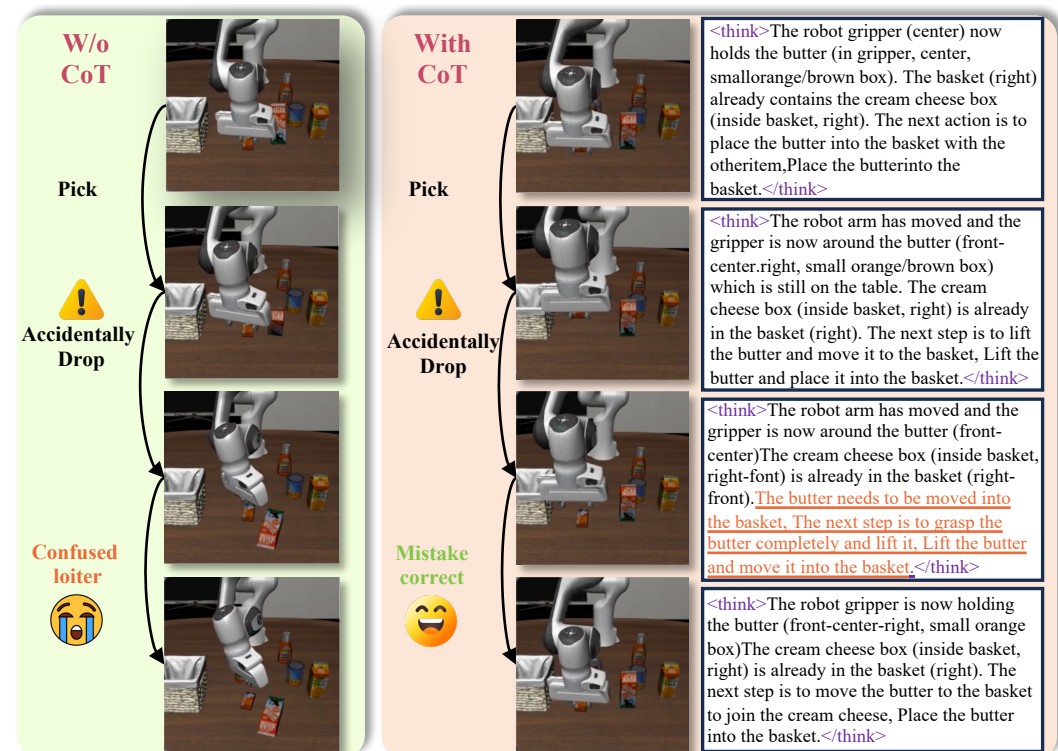

Figure 6: **"Think before acting" enables error recovery.** Comparison of rollouts on a LIBERO task. **Left:** the baseline misses the grasp and falls into a repetitive failure loop. **Right:** Deep-ThinkVLA leverages a reasoning trace to restate the subgoal, correct mistakes, and complete the task.

## A.6 Detailed Ablation: SFT vs. RL on RoboTwin 2.0

To substantiate the benefits of RL in high-fidelity, long-horizon settings, we provide the detailed breakdown of performance on the RoboTwin 2.0 benchmark in Table 5. The data confirms that RL provides consistent gains across all task complexities, with the most significant improvements observed in medium (+9.5%) and short (+7.5%) horizons, validating the effectiveness of outcome-based alignment.

## A.7 LLM Usage Statement

During the preparation of this manuscript, the authors utilized a large language model to assist with aspects of writing, including grammar correction, rephrasing for clarity, and improving the overall prose. In accordance with ICLR 2026 policy, we confirm that all authors have carefully reviewed and edited the final text and take full responsibility for its scientific accuracy, claims, and conclusions. Beyond language polishing, we also used a cloud-hosted LVLM to generate keyframe-level chain-of-thought annotations during dataset construction; all LVLM outputs were automatically schema-checked and consistency-filtered, with manual spot audits before use.

Table 5: **Detailed Ablation of RL vs. SFT on RoboTwin 2.0.** The comparison demonstrates that the RL stage significantly boosts performance across all task horizons compared to the SFT-only baseline, yielding an overall improvement of **+6.8%** (from 47.5% to 54.3%). This validates that outcome-based alignment is essential for handling complex dynamics.

| Short Horizon Tasks (100–130 Steps) | | | | |
|---|---|---|---|---|
| **Model** | **Lift Pot** | **Beat Hammer Block** | **Pick Dual Bottles** | **Place Phone Stand** | **Avg** |
| DeepThinkVLA(SFT) | 61.0 | 64.0 | 46.0 | 19.0 | 47.5 |
| DeepThinkVLA(RL) | 62.0 | 73.0 | 61.0 | 24.0 | 55.0 |
| $\Delta$ | +1.0 | +9.0 | +15.0 | +5.0 | +7.5 |
| **Medium Horizon Tasks (150–230 Steps)** | | | | |
| **Model** | **Move Can Pot** | **Place A2B Left** | **Place Empty Cup** | **Handover Mic** | **Avg** |
| DeepThinkVLA(SFT) | 33.0 | 30.0 | 78.0 | 82.0 | 55.8 |
| DeepThinkVLA(RL) | 52.0 | 38.0 | 83.0 | 88.0 | 65.3 |
| $\Delta$ | +19.0 | +8.0 | +5.0 | +6.0 | +9.5 |
| **Long (280–320 Steps) & Extra Long Horizon Tasks (450–650 Steps)** | | | | |
| **Model** | **Handover Block** | **Stack Bowls Two** | **Blocks Rank Rgb** | **Put Bottles Dustbin** | **Avg** |
| DeepThinkVLA(SFT) | 40.0 | 59.0 | 76.0 | 42.0 | 54.3 |
| DeepThinkVLA(RL) | 43.0 | 62.0 | 77.0 | 49.0 | 57.8 |
| $\Delta$ | +3.0 | +3.0 | +1.0 | +7.0 | +3.5 |
| **Overall Gap ($\Delta$ = RL - SFT): +6.8** | | | | |

