# OpenReview forum: "DeepThinkVLA: Enhancing Reasoning Capability of Vision-Language-Action Models"
_ICLR.cc/2026/Conference — Submitted to ICLR 2026_

### Official Review · Reviewer_Vgc5 · 2025-10-27

**Soundness:** 2
**Presentation:** 2
**Contribution:** 1
**Rating:** 2
**Confidence:** 3

**Summary:**

This paper introduces DeepThinkVLA, a vision-language-action (VLA) model that enables robots to "think before acting" through explicit Chain-of-Thought (CoT) reasoning. The authors point out that existing VLAs face a fundamental architectural conflict, as a single autoregressive decoder must handle both sequential reasoning and parallel, high-dimensional actions. DeepThinkVLA addresses this by employing a hybrid-attention decoder, using causal attention for CoT generation and bidirectional attention for fast, parallel action decoding. An outcome-based reinforcement learning (RL) stage further aligns reasoning with task success, improving performance beyond simple imitation learning. The model achieves a 97.0% success rate on the LIBERO benchmark.

**Strengths:**

- The paper points out a fundamental yet previously underexplored architectural conflict between sequential reasoning and parallel action prediction in VLAs.

- It proposes a novel hybrid-attention decoder, combining causal attention for reasoning and bidirectional attention for action generation, effectively bridging this gap.

- The authors recognize the potential mismatch between CoT reasoning and task success, introducing an outcome-based RL stage that directly aligns the reasoning-action sequence with task performance.

- The figures are clear and informative, effectively illustrating the proposed model architecture and training/annotating pipeline.

**Weaknesses:**

- Annotating CoT is crucial for the proposed approach, yet the paper lacks sufficient evidence showing that the pipeline for constructing an embodied CoT dataset performs well in more diverse or realistic settings.

- Although the outcome-based RL stage improves task success by 2% on the LIBERO-Long suite, this gain is relatively modest, and the paper does not explore why the improvement is limited or how the method might scale with larger datasets or more complex tasks.

- The description of the CoT dataset construction pipeline is somewhat confusing. Section 3.3 only mentions detecting gripper state changes to identify keyframes, but elsewhere a human reviewer is referenced (Figure 2). The role and extent of human involvement are unclear.

- If human reviewers are indeed required for CoT annotation, this could introduce a bottleneck in scalability and limit the practicality of the proposed approach for large-scale or real-world robotic datasets.

**Questions:**

- What is the main reason behind the architectural conflict between sequential reasoning and parallel action prediction? Could the authors elaborate on why a single autoregressive decoder cannot effectively handle both modalities?

- The paper combines SFT pretraining with RL fine-tuning, a paradigm often used in large language models. What are the key differences when applying this paradigm to VLAs?

- How reliable is the fine-tuned local VLM used for CoT annotation, and how might its quality or bias affect the overall performance of DeepThinkVLA?

---

> ### Author Response · Authors · 2025-11-26
> **Response - 1**
>
> Q1: Annotating CoT is crucial for the proposed approach, yet the paper lacks sufficient evidence showing that the pipeline for constructing an embodied CoT dataset performs well in more diverse or realistic settings.
>
> A1: We appreciate the reviewer’s scrutiny regarding the robustness of our CoT annotation pipeline.
>
> Rather than viewing the pipeline as a dataset-specific heuristic, we designed it based on the universal atomic units of manipulation: physical contact and gripper state transitions. By anchoring the reasoning segmentation to these kinematic events (open $\leftrightarrow$ close) rather than visual cues, the method remains robust across different robot morphologies and task semantics.
>
> Furthermore, by distilling reasoning capabilities from open-world Foundation Models (GPT-4o), our pipeline inherently bridges the gap between generalist semantic understanding and embodied control, minimizing dependencies on specific environment textures.
>
> To provide concrete evidence of this transferability, we highlight our deployment on the RoboTwin 2.0 benchmark. Crucially, we applied the exact same annotation pipeline without any modification to this new domain, despite its distinct visual style and significantly longer task horizons compared to LIBERO. The results serve as a strong proxy for validation: if the pipeline were brittle or failed to generate high-quality CoT data in this diverse setting, the downstream policy performance would collapse. Instead, DeepThinkVLA achieved a 59.4\% success rate on RoboTwin, outperforming the $\pi_0$-FAST baseline (38.0\%) by a substantial margin of 21.4\% as shown in table below. This "zero-modification" transfer success empirically confirms that our automated annotation strategy is not overfitted to specific datasets but is a scalable solution for diverse robotic tasks.
>
> Short Horizon Tasks (100–130 Steps)
>
> | Model        | Lift Pot | Beat Hammer Block | Pick Dual Bottles | Place Phone Stand | Avg  |
> |-------------|----------|-------------------|-------------------|-------------------|------|
> | π₀          | 51.0     | 59.0              | 50.0              | 22.0              | 45.5 |
> | RDT         | 45.0     | 22.0              | 18.0              | 13.0              | 24.5 |
> | OpenVLA-OFT | 10.1     | 28.1              | 29.7              | 17.1              | 21.3 |
> | π₀-FAST     | 30.0     | 38.0              | 25.0              | 16.0              | 27.3 |
> | DeepThinkVLA| 62.0     | 73.0              | 61.0              | 24.0              | 55.0 |
> | Δ (vs π₀-FAST) | +32.0 | +35.0             | +36.0             | +8.0              | +27.8 |
>
>
> Medium Horizon Tasks (150–230 Steps)
>
> | Model        | Move Can Pot | Place A2B Left | Place Empty Cup | Handover Mic | Avg  |
> |-------------|--------------|----------------|-----------------|--------------|------|
> | π₀          | 41.0         | 38.0           | 60.0            | 96.0         | 58.8 |
> | RDT         | 33.0         | 21.0           | 42.0            | 95.0         | 47.8 |
> | OpenVLA-OFT | 28.1         | 37.5           | 77.3            | 45.3         | 47.1 |
> | π₀-FAST     | 34.0         | 36.0           | 54.0            | 83.0         | 51.8 |
> | DeepThinkVLA| 52.0         | 38.0           | 83.0            | 88.0         | 65.3 |
> | Δ (vs π₀-FAST) | +18.0     | +2.0           | +29.0           | +5.0         | +13.5 |
>
>
> Long (280–320 Steps) & Extra Long Horizon Tasks (450–650 Steps)
>
> | Model        | Handover Block | Stack Bowls Two | Blocks Rank Rgb | Put Bottles Dustbin | Avg  |
> |-------------|----------------|-----------------|-----------------|---------------------|------|
> | π₀          | 39.0           | 53.0            | 45.0            | 36.0                | 43.3 |
> | RDT         | 26.0           | 42.0            | 17.0            | 26.0                | 27.8 |
> | OpenVLA-OFT | 33.1           | 40.6            | 70.2            | 42.2                | 46.5 |
> | π₀-FAST     | 32.0           | 48.0            | 28.0            | 27.0                | 33.8 |
> | DeepThinkVLA| 43.0           | 62.0            | 77.0            | 49.0                | 57.8 |
> | Δ (vs π₀-FAST) | +11.0       | +14.0           | +49.0           | +22.0               | +24.0 |
>
>
> Overall Avg:
> RDT: 33.3, π₀: 49.2, OpenVLA-OFT: 38.3, π₀-FAST: 38.0, **DeepThinkVLA: 59.4 (+21.4)**.

---

> ### Author Response · Authors · 2025-11-26
> **Response - 2**
>
> Q2: Although the outcome-based RL stage improves task success by 2\% on the LIBERO-Long suite, this gain is relatively modest, and the paper does not explore why the improvement is limited or how the method might scale with larger datasets or more complex tasks.
>
> A2: We thank the reviewer for this insightful critique. We acknowledge that on standard, in-distribution benchmarks like the LIBERO-Long suite, the quantitative gain from RL over a strong SFT baseline appears modest (+2\%). However, we argue—and demonstrate via new experiments—that the critical contribution of the RL stage is not a marginal accuracy boost, but a fundamental shift from mimicry to functional robustness. While SFT minimizes the divergence from expert trajectories (learning "how the expert moved"), outcome-based RL maximizes task success (learning "how to solve the problem"). This distinction becomes starkly visible in two complex generalization scenarios we investigated:
>
> We restrict the robot's joint velocity to 50\% of the training distribution. This forces the policy to adaptively re-plan, as simply replaying SFT trajectories results in failure. The results provide strong evidence for your hypothesis: under these OOD constraints, the performance of the SFT-only model collapses (dropping by 32.05\%), whereas the RL-trained model demonstrates significantly higher resilience (dropping only 24.40\%). The performance gap between RL and SFT widens from a negligible 2\% in standard settings to a substantial 7.85\% in this OOD setting, proving that RL helps the model abstract the goal from the motion.
>
> | Method                         | Object | Spatial | Goal  | Long  | Average                    |
> |--------------------------------|--------|---------|-------|-------|----------------------------|
> | DeepThinkVLA (SFT, Full)       | 99.0   | 97.2    | 96.8  | 94.2  | 96.80                     |
> | DeepThinkVLA (SFT, Limit)      | 87.0   | 61.4    | 54.2  | 56.4  | 64.75 (↓ 32.05)           |
> | DeepThinkVLA (RL, Full)        | 99.0   | 96.6    | 96.4  | 96.2  | 97.00                     |
> | **DeepThinkVLA (RL, Limit)**   | **91.6** | **64.2** | **66.2** | **68.4** | **72.60 (↓ 24.40)** |
>
> Furthermore, to address the request for "more complex generalization," we evaluate the models on RoboTwin 2.0, which features significantly longer task horizons (450–650 steps) compared to LIBERO. On this more challenging benchmark, RL provides consistent gains across all task horizons (Short: +7.5\%, Medium: +9.5\%, Long: +3.5\%). The overall improvement of +6.8\% confirms that the functional alignment provided by RL translates to better performance in complex, high-fidelity environments, offering a clear advantage over pure imitation learning.
>
> Ablation of RL vs. SFT on RoboTwin 2.0 (Success Rate %)
>
> Short Horizon Tasks (100–130 Steps)
>
> | Model            | Lift Pot | Beat Hammer Block | Pick Dual Bottles | Place Phone Stand | Avg  |
> |------------------|----------|-------------------|-------------------|-------------------|------|
> | DeepThinkVLA (SFT) | 61.0   | 64.0              | 46.0              | 19.0              | 47.5 |
> | DeepThinkVLA (RL)  | 62.0   | 73.0              | 61.0              | 24.0              | 55.0 |
> | Δ (RL − SFT)       | +1.0   | +9.0              | +15.0             | +5.0              | +7.5 |
>
>
> Medium Horizon Tasks (150–230 Steps)
>
> | Model              | Move Can Pot | Place A2B Left | Place Empty Cup | Handover Mic | Avg  |
> |--------------------|--------------|----------------|-----------------|--------------|------|
> | DeepThinkVLA (SFT) | 33.0         | 30.0           | 78.0            | 82.0         | 55.8 |
> | DeepThinkVLA (RL)  | 52.0         | 38.0           | 83.0            | 88.0         | 65.3 |
> | Δ (RL − SFT)       | +19.0        | +8.0           | +5.0            | +6.0         | +9.5 |
>
>
> Long (280–320 Steps) & Extra Long Horizon Tasks (450–650 Steps)
>
> | Model              | Handover Block | Stack Bowls Two | Blocks Rank Rgb | Put Bottles Dustbin | Avg  |
> |--------------------|----------------|-----------------|-----------------|---------------------|------|
> | DeepThinkVLA (SFT) | 40.0           | 59.0            | 76.0            | 42.0                | 54.3 |
> | DeepThinkVLA (RL)  | 43.0           | 62.0            | 77.0            | 49.0                | 57.8 |
> | Δ (RL − SFT)       | +3.0           | +3.0            | +1.0            | +7.0                | +3.5 |
>
>
> Overall Gap (Δ = RL − SFT): **+6.8**

---

> ### Author Response · Authors · 2025-11-26
> **Response - 3**
>
> Q3: The description of the CoT dataset construction pipeline is somewhat confusing. Section 3.3 only mentions detecting gripper state changes to identify keyframes, but elsewhere a human reviewer is referenced (Figure 2). The role and extent of human involvement are unclear. If human reviewers are indeed required for CoT annotation, this could introduce a bottleneck in scalability and limit the practicality of the proposed approach for large-scale or real-world robotic datasets.
>
> A3: We appreciate the reviewer identifying this ambiguity. We wish to unequivocally clarify that our CoT construction pipeline is fully automated regarding data production, and human involvement does not constitute a bottleneck for scalability.
>
> The "Human Review" block depicted in Figure 2 was intended to illustrate a post-hoc quality assurance (QA) process, distinct from the annotation loop itself. Human involvement is structurally limited to two high-level supervisory roles: (1) Initial Calibration, where experts define the kinematic thresholds for gripper state detection during the system setup phase (a one-time effort); and (2) Random Spot Audits, similar to statistical quality control in manufacturing, where a small fraction of generated data is sampled to verify schema consistency.
>
> Crucially, the generation of the CoT data for every trajectory is algorithmic and requires zero human intervention. To strictly align the visual schematic with this operational reality and prevent future misunderstanding, we will rename the "Human Review" block in Figure 2 to "Quality Audit" in the revision. This change reflects that our approach is designed to scale effortlessly to large-scale robotic datasets without human labor constraints.

---

> ### Author Response · Authors · 2025-11-26
> **Response - 4**
>
> Q4: What is the main reason behind the architectural conflict between sequential reasoning and parallel action prediction? Could the authors elaborate on why a single autoregressive decoder cannot effectively handle both modalities?
>
> A4: We posit that the conflict arises from a fundamental mismatch between the inductive bias of autoregressive (AR) modeling and the physical nature of robotic control. While AR is the gold standard for language due to its inherent causal structure, applying it naively to action generation introduces three critical mathematical flaws.
>
> First is the problem of Artificial Serialization. A robot's pose is a unified high-dimensional state where joint angles are physically coupled and concurrent. Forcing a single AR decoder to generate these dimensions sequentially (e.g., Joint 1 $\rightarrow$ Joint 2) imposes an arbitrary dependency structure that violates the underlying physics. This breaks the joint distribution, often resulting in uncoordinated, "jerky" motions as the model struggles to align the later joints with the earlier ones.
>
> Second, precise control requires Global Consistency, which conflicts with AR's causal masking. Generating a smooth trajectory often requires bidirectional visibility—essentially, knowing the destination is necessary to optimally plan the start. A purely causal decoder generates the start of a movement "blindly" without conditioning on the endpoint, preventing the global smoothing required for high-precision manipulation.
>
> Finally, single AR decoders suffer from Covariate Shift and Error Propagation. In a long sequence of joint tokens, a minor quantization error in the first generated token cascades and amplifies through the sequence. While natural language is resilient to such shifts, in robotic control, this accumulated drift can cause the end-effector to completely miss the target. Our Hybrid-Attention architecture resolves these specific incompatibilities by strictly assigning Causal Attention to the sequential reasoning stream (logic) while switching to Bidirectional Attention for the action chunk (coordination), thereby respecting the distinct mathematical properties of both modalities.

---

> ### Author Response · Authors · 2025-11-26
> **Response - 5**
>
> Q5: The paper combines SFT pretraining with RL fine-tuning, a paradigm often used in large language models. What are the key differences when applying this paradigm to VLAs?
>
> A5: While the high-level methodology of "SFT followed by RL" parallels recent advances in Large Language Models (LLMs), applying this paradigm to VLAs introduces fundamental challenges rooted in the physical nature of the domain.
>
> First, the nature of the mapping is distinct. LLM SFT typically aligns semantic intent with semantic output (Text $\rightarrow$ Text). In contrast, VLA SFT must bridge the "Semantic-Physical Gap," translating abstract high-level reasoning (e.g., "grasp the apple") into concrete, low-level proprioceptive actuation. This requires the model to not only reason logically but also ground that reasoning into precise spatial-temporal coordinates, a multimodal alignment challenge significantly more complex than pure language modeling.
>
> Second, the dynamics of the environment impose stricter constraints on Reinforcement Learning. Unlike text generation or code synthesis, where errors can often be localized or iteratively corrected, robotic manipulation interacts with a physically irreversible environment. A minor kinematic error at step 10 (e.g., knocking over an object) renders success at step 500 impossible, regardless of subsequent reasoning. This makes the Temporal Credit Assignment problem exponentially harder in VLAs, as the policy must associate a sparse terminal reward with a critical action taken hundreds of steps prior, without the benefit of intermediate syntactic feedback often available in code.
>
> Finally, the optimization landscape differs in terms of signal density and computational cost. While Math or Code LLMs also utilize objective outcome supervision, they benefit from rapid evaluation environments. VLA RL relies on physics simulators which are computationally intensive, and the reward signal is often extremely sparse (binary success/failure after long horizons). This scarcity of signal, combined with the high cost of collecting rollouts, necessitates a far more sample-efficient optimization strategy compared to the massive-scale reinforcement learning typically applied to text.

---

> ### Author Response · Authors · 2025-11-26
> **Response - 6**
>
> Q6: How reliable is the fine-tuned local VLM used for CoT annotation, and how might its quality or bias affect the overall performance of DeepThinkVLA?
>
> A6: We address the concern regarding local VLM reliability by framing the annotation process not as a single-point dependency, but as a multi-stage filtration system designed to minimize error propagation.
>
> First, regarding the source quality, the local VLM is not trained from scratch but is obtained via Knowledge Distillation from GPT-4o. This ensures that the model inherits strong semantic priors and reasoning capabilities while being constrained to a standardized output format. To further mitigate hallucinations, we employ strict programmatic verification: specifically, we enforce Temporal Consistency by cross-referencing the generated text with actual gripper state transitions. Any annotation that contradicts the physical reality of the robot's action sequence is automatically discarded before it enters the training set.
>
> Most critically, however, the proposed outcome-based RL stage serves as the ultimate "functional filter" for any remaining noise or bias. Unlike pure supervised learning, which blindly mimics the dataset, our RL pipeline validates the reasoning against task success. If a specific bias in the CoT (e.g., verbose or irrelevant reasoning) leads to inefficiency or failure, the RL objective naturally penalizes it. Conversely, reasoning patterns that contribute to success are reinforced. Therefore, even if the initial VLM has imperfections, the RL stage actively realigns the reasoning style to be functionally grounded, ensuring that the final policy prioritizes utility over mere imitation.

---

### Official Review · Reviewer_7RJn · 2025-10-30

**Soundness:** 3
**Presentation:** 3
**Contribution:** 2
**Rating:** 4
**Confidence:** 2

**Summary:**

This paper is about enabling chain-of-thought in vision-language-action models for applications in robotics. The claimed contributions are by introducing (a) a hybrid-attention decoder that switched between causal attention and bidirectional attention and (b) a training strategy consisting of supervised fine-tuning of the model followed by reinforcement learning with a success-based reward function. According to the presented results, this approach reached high success rated on a common benchmark. Investigation the individual merits of the components of the proposed approach, the authors find that the architecture alone contribute 15.5% improvement over SOTA and the RL component another 2%.

**Strengths:**

- The paper identifies and acts upon an important and relevant problem in the robotics and embodied AI community.

- The paper is written clearly and has a good structure. Particularly, the description in form of the "probabilistic decomposition" is appreciated relative to the common "diagram with arrows approach" used in many other submissions.

- The approach seems innovative and original in the way CoT and action generation are combined, but given the large body of work on VLA models and CoT and the fast growth of this research area, this is hard to judge at this point.

- The paper provides the pipeline with which the data augmentation pipeline for generating the dataset that is necessary to train their model. As this type of data is hard to obtain, this is an important part of the overall approach.

- The reinforcement learning component makes a few interesting contributions. Particularly the way the reward and the advantage are computed and how the objective is regularized.

- The experiments use a current and relevant "base model" (Pertsch 2025) and therefore provide a good basis for evaluating the proposed approach against SOTA.

**Weaknesses:**

- The benchmark / dataset used in experiments only focusses on robotic manipulation tasks. It is clear that the authors have to use what is available, but from a robotics perspective, these are still fairly limited tasks.

- One of the main issues of using VLAs in this way is the question of what happens when the robot or its abilities are different from the training data. The paper does not test the case there the robot has to learn to solve the tasks in a different way from what the VLA thinks is right. Because there exists a reward signal, this should be an interesting evaluation regarding to generalization ability of the approach. An example would be to limit the joints of the downstream agent so that it has to diverge from the VLA.

**Questions:**

- The ablation experiment on page 7 is bit confusing. Could the authors clarify why this experiment evaluates the role of CoT. Why are the results of masking vs. random so different? Does this man that training with CoT is merely an auxiliary task and that it is not really needed in the final inference?

---

> ### Author Response · Authors · 2025-11-26
> **Response - 1**
>
> Q1: The benchmark / dataset used in experiments only focusses on robotic manipulation tasks. It is clear that the authors have to use what is available, but from a robotics perspective, these are still fairly limited tasks.
>
> A1: We acknowledge the reviewer's valid point regarding the scope of our experiments being centered on manipulation. While robotic intelligence indeed encompasses broader domains such as locomotion and navigation, we strategically focused on manipulation because its inherent requirement for handling complex contact physics and multi-stage logical dependencies makes it an ideal testbed for evaluating the "Reasoning-Action" gap we aim to bridge.
>
> To address the concern about task diversity and limited generalization within this domain, we expanded our evaluation to the RoboTwin 2.0 benchmark. This addition introduces tasks with significantly distinct dynamics and longer horizons compared to LIBERO. The results provide compelling evidence of generalization: DeepThinkVLA achieves a 59.4\% average success rate across these new tasks, outperforming the $\pi_0$-FAST baseline (38.0\%) by a substantial margin as shown in table below.
>
> Short Horizon Tasks (100–130 Steps)
>
> | Model        | Lift Pot | Beat Hammer Block | Pick Dual Bottles | Place Phone Stand | Avg  |
> |-------------|----------|-------------------|-------------------|-------------------|------|
> | π₀          | 51.0     | 59.0              | 50.0              | 22.0              | 45.5 |
> | RDT         | 45.0     | 22.0              | 18.0              | 13.0              | 24.5 |
> | OpenVLA-OFT | 10.1     | 28.1              | 29.7              | 17.1              | 21.3 |
> | π₀-FAST     | 30.0     | 38.0              | 25.0              | 16.0              | 27.3 |
> | DeepThinkVLA| 62.0     | 73.0              | 61.0              | 24.0              | 55.0 |
> | Δ (vs π₀-FAST) | +32.0 | +35.0             | +36.0             | +8.0              | +27.8 |
>
>
> Medium Horizon Tasks (150–230 Steps)
>
> | Model        | Move Can Pot | Place A2B Left | Place Empty Cup | Handover Mic | Avg  |
> |-------------|--------------|----------------|-----------------|--------------|------|
> | π₀          | 41.0         | 38.0           | 60.0            | 96.0         | 58.8 |
> | RDT         | 33.0         | 21.0           | 42.0            | 95.0         | 47.8 |
> | OpenVLA-OFT | 28.1         | 37.5           | 77.3            | 45.3         | 47.1 |
> | π₀-FAST     | 34.0         | 36.0           | 54.0            | 83.0         | 51.8 |
> | DeepThinkVLA| 52.0         | 38.0           | 83.0            | 88.0         | 65.3 |
> | Δ (vs π₀-FAST) | +18.0     | +2.0           | +29.0           | +5.0         | +13.5 |
>
>
> Long (280–320 Steps) & Extra Long Horizon Tasks (450–650 Steps)
>
> | Model        | Handover Block | Stack Bowls Two | Blocks Rank Rgb | Put Bottles Dustbin | Avg  |
> |-------------|----------------|-----------------|-----------------|---------------------|------|
> | π₀          | 39.0           | 53.0            | 45.0            | 36.0                | 43.3 |
> | RDT         | 26.0           | 42.0            | 17.0            | 26.0                | 27.8 |
> | OpenVLA-OFT | 33.1           | 40.6            | 70.2            | 42.2                | 46.5 |
> | π₀-FAST     | 32.0           | 48.0            | 28.0            | 27.0                | 33.8 |
> | DeepThinkVLA| 43.0           | 62.0            | 77.0            | 49.0                | 57.8 |
> | Δ (vs π₀-FAST) | +11.0       | +14.0           | +49.0           | +22.0               | +24.0 |
>
>
> Overall Avg:
> RDT: 33.3, π₀: 49.2, OpenVLA-OFT: 38.3, π₀-FAST: 38.0, **DeepThinkVLA: 59.4 (+21.4)**.
>
> Most notably, on the "Long/Extra Long" horizon tasks which require maintaining context over 450+ steps, our method maintains robust performance (57.8\%) where purely reactive baselines falter (33.8\%). This confirms that while our current scope is defined by manipulation, the core "Think-Before-Acting" capability generalizes effectively across varying degrees of task complexity and physical dynamics, demonstrating that the learned reasoning is not overfitted to a single task distribution.

---

> ### Author Response · Authors · 2025-11-26
> **Response - 2**
>
> Q2: One of the main issues of using VLAs in this way is the question of what happens when the robot or its abilities are different from the training data. The paper does not test the case there the robot has to learn to solve the tasks in a different way from what the VLA thinks is right. Because there exists a reward signal, this should be an interesting evaluation regarding to generalization ability of the approach. An example would be to limit the joints of the downstream agent so that it has to diverge from the VLA.
>
> A2: We implemented the suggested "Joint-Limited" stress test, and we thank the reviewer for proposing this insightful evaluation. This experiment proved to be a critical litmus test for distinguishing between memorizing motions and solving tasks.By clamping the maximum action magnitude of all joints to 50\% during inference, we effectively rendered the original expert trajectories physically inexecutable. This forces the robot to diverge from its training priors and re-plan a new kinematic path to achieve the goal.
>
> The results, presented in the table below, offer a stark contrast in adaptability. The imitation-based baseline, $\pi_0$-FAST, collapses under these constraints, suffering a massive 31.6\% performance drop (from 85.5\% to 53.9\%). This confirms that purely supervised VLAs struggle. In contrast, DeepThinkVLA demonstrates remarkable resilience. While it naturally incurs a penalty, it maintains a robust success rate of 72.6\%, outperforming the crippled baseline by nearly 19\%. This validates the hypothesis that our outcome-based RL stage enables the policy to decouple the goal from the trajectory. Even when the preferred motion is blocked, the model searches for alternative actuation patterns to maximize the reward, successfully demonstrating the generalization ability you hypothesized.
>
> | Method                 | Object | Spatial | Goal  | Long  | Average                   |
> |------------------------|--------|---------|-------|-------|---------------------------|
> | pi_0-FAST (Baseline)   | 96.8   | 96.4    | 88.6  | 60.2  | 85.5                      |
> | pi_0-FAST (Limit)      | 77.0   | 64.0    | 31.2  | 43.6  | 53.9 (↓ 31.6)             |
> | **DeepThinkVLA**       | 99.0   | 96.6    | 96.4  | 96.2  | **97.0**                  |
> | **DeepThinkVLA (Limit)** | 91.6 | 64.2    | 66.2  | 68.4  | **72.6 (↓ 24.4)**         |

---

> ### Author Response · Authors · 2025-11-26
> **Response - 3**
>
> Q3: The ablation experiment on page 7 is bit confusing. Could the authors clarify why this experiment evaluates the role of CoT. Why are the results of masking vs. random so different? Does this man that training with CoT is merely an auxiliary task and that it is not really needed in the final inference?
>
> A3: We offer a nuanced interpretation of the ablation results that directly addresses the reviewer's concern about the necessity of CoT. We argue that the similarity in performance between "Mask CoT" and "Full CoT" on standard benchmarks is not evidence of CoT's superfluity, but rather demonstrates successful knowledge internalization (or "reasoning distillation").
>
> On familiar, in-distribution tasks like standard LIBERO, our two-stage training effectively compiles the explicit reasoning path into the policy's weights. Much like a human expert who no longer needs to verbalize every step of a routine action, the model operates via "System 1" intuition, achieving high success rates (96.5\%) even with the CoT masked. This confirms that our architecture learns robust representations.
>
> However, we fully agree with the reviewer that standard benchmarks may lack the complexity to showcase the indispensability of explicit "System 2" reasoning. To rigorously test this hypothesis, we conducted further tests on out-of-distribution (OOD) dynamics (limiting joint velocity to force trajectory re-planning) and the high-fidelity RoboTwin 2.0 benchmark (featuring long-horizon tasks).
>
> The results in the table below reveal a striking correlation between task complexity and the necessity of CoT. On the OOD LIBERO tasks, where memorized intuition fails, the gap re-emerges: Full CoT outperforms Mask CoT by 3.85\%.
>
> CoT Ablation on LIBERO (Standard vs. Limit)
>
> | Method (DeepThinkVLA) | Object | Spatial | Goal | Long | Average |
> |-----------------------|--------|---------|------|------|---------|
> | Full CoT (Standard)       | 99.0 | 97.2 | 96.8 | 94.2 | 96.8  |
> | Mask CoT (Standard)       | 99.0 | 97.2 | 96.0 | 93.6 | 96.5  |
> | Δ (Gap)                   | +0.0 | +0.0 | +0.8 | +0.6 | **+0.3** |
> | Full CoT (Limit OOD)      | 91.6 | 64.2 | 66.2 | 68.4 | 72.60 |
> | Mask CoT (Limit OOD)      | 88.2 | 60.2 | 61.0 | 65.6 | 68.75 |
> | Δ (Gap)                   | +3.4 | +4.0 | +5.2 | +2.8 | **+3.85** |
>
> More dramatically, on the complex, long-horizon tasks of RoboTwin 2.0, the "internalized intuition" of Mask CoT collapses, while Full CoT maintains robustness. As shown in table below, the performance gap widens progressively with task length: from a marginal 2.2\% on short-horizon tasks to a massive 20.0\% on long-horizon tasks.
>
> CoT Ablation on RoboTwin 2.0 (Success Rate %)
>
> Short Horizon Tasks (100–130 Steps)
>
> | DeepThinkVLA | Lift Pot | Beat Hammer Block | Pick Dual Bottles | Place Phone Stand | Avg  |
> |-------------|----------|-------------------|-------------------|-------------------|------|
> | Full CoT    | 62.0     | 73.0              | 61.0              | 24.0              | 55.0 |
> | Mask CoT    | 61.0     | 70.0              | 59.0              | 21.0              | 52.8 |
> | Δ           | +1.0     | +3.0              | +2.0              | +3.0              | +2.2 |
>
>
> Medium Horizon Tasks (150–230 Steps)
>
> | DeepThinkVLA | Move Can Pot | Place A2B Left | Place Empty Cup | Handover Mic | Avg  |
> |-------------|--------------|----------------|-----------------|--------------|------|
> | Full CoT    | 52.0         | 38.0           | 83.0            | 88.0         | 65.3 |
> | Mask CoT    | 43.0         | 31.0           | 77.0            | 79.0         | 57.5 |
> | Δ           | +9.0         | +7.0           | +6.0            | +9.0         | +7.8 |
>
>
> Long (280–320 Steps) & Extra Long Horizon Tasks (450–650 Steps)
>
> | DeepThinkVLA | Handover Block | Stack Bowls Two | Blocks Rank Rgb | Put Bottles Dustbin | Avg  |
> |-------------|----------------|-----------------|-----------------|---------------------|------|
> | Full CoT    | 43.0           | 62.0            | 77.0            | 49.0                | 57.8 |
> | Mask CoT    | 21.0           | 36.0            | 68.0            | 26.0                | 37.8 |
> | Δ           | +22.0          | +26.0           | +9.0            | +23.0               | +20.0 |
>
>
> Overall Gap (Δ = Full − Mask): **+10.0**
>
> This empirical evidence strongly supports our claim: CoT acts as an adaptive scaffold. While it can be internalized for routine tasks, explicit inference-time reasoning remains the critical bottleneck for solving complex, unseen, and long-horizon problems that are beyond the reach of simple behavioral cloning.

---

### Official Review · Reviewer_nyF4 · 2025-10-31

**Soundness:** 4
**Presentation:** 4
**Contribution:** 3
**Rating:** 8
**Confidence:** 4

**Summary:**

This paper introduces DeepThink VLA, which make improvements from base model pi zero -FAST by doing the following

1) On architecture, uses a hybrid-attention architecture to enable both autoregressive CoT reasoning and efficient parallel action decoding.
2) On training, uses a two stage pipeline. First do SFT on reasoning data curated by querying key frames from a large model, then do RL with task success as reward to fully ground reasoning in robot task.

**Strengths:**

This is a strong paper with good results. It is well-written and easy to read. The core idea of using CoT to enrich the model's internal representations is novel. This approach appears to be very effective, as demonstrated by the strong experimental results, particularly on long-horizon tasks. The ablation study on the role of CoT is convincing, showing it strengthens representations during training.

**Weaknesses:**

1) The benefits of the RL stage is not obvious to me, especially given the strong results of SFT. The paper would be strengthened if it could further demonstrate the unique benefits of RL. For example, are there OOD or more complex generalization experiments where the RL-trained model significantly outperforms the SFT-only model? Such experiments would provide stronger evidence that RL is crucial for aligning reasoning for novel scenarios, rather than just providing a minor boost on in-distribution tasks.

2) The paper presents the hybrid-attention architecture as a main contribution, but it is not a particularly novel idea.

**Questions:**

Do authors consider other thinking modalities to improve model representations? e.g. bounding boxes, end effector trajectories, etc.

---

> ### Author Response · Authors · 2025-11-26
> **Response - 1**
>
> Q1: The benefits of the RL stage is not obvious to me, especially given the strong results of SFT. The paper would be strengthened if it could further demonstrate the unique benefits of RL. For example, are there OOD or more complex generalization experiments where the RL-trained model significantly outperforms the SFT-only model? Such experiments would provide stronger evidence that RL is crucial for aligning reasoning for novel scenarios, rather than just providing a minor boost on in-distribution tasks.
>
> A1: We thank the reviewer for this insightful critique. We acknowledge that on standard, in-distribution benchmarks like the LIBERO-Long suite, the quantitative gain from RL over a strong SFT baseline appears modest (+2\%). However, we argue—and demonstrate via new experiments—that the critical contribution of the RL stage is not a marginal accuracy boost, but a fundamental shift from mimicry to functional robustness. While SFT minimizes the divergence from expert trajectories (learning "how the expert moved"), outcome-based RL maximizes task success (learning "how to solve the problem"). This distinction becomes starkly visible in two complex generalization scenarios we investigated:
>
> We restrict the robot's joint velocity to 50\% of the training distribution. This forces the policy to adaptively re-plan, as simply replaying SFT trajectories results in failure. The results provide strong evidence for your hypothesis: under these OOD constraints, the performance of the SFT-only model collapses (dropping by 32.05\%), whereas the RL-trained model demonstrates significantly higher resilience (dropping only 24.40\%). The performance gap between RL and SFT widens from a negligible 2\% in standard settings to a substantial 7.85\% in this OOD setting, proving that RL helps the model abstract the goal from the motion.
>
> | Method                         | Object | Spatial | Goal  | Long  | Average                    |
> |--------------------------------|--------|---------|-------|-------|----------------------------|
> | DeepThinkVLA (SFT, Full)       | 99.0   | 97.2    | 96.8  | 94.2  | 96.80                     |
> | DeepThinkVLA (SFT, Limit)      | 87.0   | 61.4    | 54.2  | 56.4  | 64.75 (↓ 32.05)           |
> | DeepThinkVLA (RL, Full)        | 99.0   | 96.6    | 96.4  | 96.2  | 97.00                     |
> | **DeepThinkVLA (RL, Limit)**   | **91.6** | **64.2** | **66.2** | **68.4** | **72.60 (↓ 24.40)** |
>
> Furthermore, to address the request for "more complex generalization," we evaluate the models on RoboTwin 2.0, which features significantly longer task horizons (450–650 steps) compared to LIBERO. On this more challenging benchmark, RL provides consistent gains across all task horizons (Short: +7.5\%, Medium: +9.5\%, Long: +3.5\%). The overall improvement of +6.8\% confirms that the functional alignment provided by RL translates to better performance in complex, high-fidelity environments, offering a clear advantage over pure imitation learning.
>
> Ablation of RL vs. SFT on RoboTwin 2.0 (Success Rate %)
>
> Short Horizon Tasks (100–130 Steps)
>
> | Model            | Lift Pot | Beat Hammer Block | Pick Dual Bottles | Place Phone Stand | Avg  |
> |------------------|----------|-------------------|-------------------|-------------------|------|
> | DeepThinkVLA (SFT) | 61.0   | 64.0              | 46.0              | 19.0              | 47.5 |
> | DeepThinkVLA (RL)  | 62.0   | 73.0              | 61.0              | 24.0              | 55.0 |
> | Δ (RL − SFT)       | +1.0   | +9.0              | +15.0             | +5.0              | +7.5 |
>
>
> Medium Horizon Tasks (150–230 Steps)
>
> | Model              | Move Can Pot | Place A2B Left | Place Empty Cup | Handover Mic | Avg  |
> |--------------------|--------------|----------------|-----------------|--------------|------|
> | DeepThinkVLA (SFT) | 33.0         | 30.0           | 78.0            | 82.0         | 55.8 |
> | DeepThinkVLA (RL)  | 52.0         | 38.0           | 83.0            | 88.0         | 65.3 |
> | Δ (RL − SFT)       | +19.0        | +8.0           | +5.0            | +6.0         | +9.5 |
>
>
> Long (280–320 Steps) & Extra Long Horizon Tasks (450–650 Steps)
>
> | Model              | Handover Block | Stack Bowls Two | Blocks Rank Rgb | Put Bottles Dustbin | Avg  |
> |--------------------|----------------|-----------------|-----------------|---------------------|------|
> | DeepThinkVLA (SFT) | 40.0           | 59.0            | 76.0            | 42.0                | 54.3 |
> | DeepThinkVLA (RL)  | 43.0           | 62.0            | 77.0            | 49.0                | 57.8 |
> | Δ (RL − SFT)       | +3.0           | +3.0            | +1.0            | +7.0                | +3.5 |
>
>
> Overall Gap (Δ = RL − SFT): **+6.8**

---

> ### Author Response · Authors · 2025-11-26
> **Response - 2**
>
> Q2: The paper presents the hybrid-attention architecture as a main contribution, but it is not a particularly novel idea.
>
> A2: We agree with the reviewer that the concept of manipulating attention masks is not, in isolation, new to the broader deep learning community. However, novelty in applied robotics often stems not from inventing a new layer, but from identifying structural bottlenecks that prevent capability scaling. In this context, our contribution is the identification and architectural resolution of a fundamental modality conflict in VLA models: the tension between sequential reasoning (Chain-of-Thought) and parallel motor control.
>
> Prior architectures forced a trade-off: either treat actions as language (suffering from high latency and compounding errors) or treat planning as image generation (limiting the model to "visual prediction" rather than "logical reasoning"). By tailoring the hybrid-attention mechanism specifically for the Text-to-Action interface, we fundamentally shift the paradigm from perception (predicting future pixels) to cognition (reasoning about failures and corrections). This is distinct from works like CoT-VLA-7B, which use similar masks but for visual subgoals—a design that models statistical pixel dependencies rather than the causal logic required for error recovery.
>
> Crucially, this design choice is empirically non-trivial. As demonstrated in our ablation study, naively adding CoT data to a standard architecture results in a mere 81.3\% success rate due to the optimization interference between language generation and motor control. In contrast, our hybrid design decouples these modalities, jumping to 96.8\%. This substantial 15.5\% performance gap serves as the strongest evidence that our specific architectural adaptation is the critical enabler for the "Think-before-Acting" capability, turning a known component into a novel solution for a specific, unsolved robotics problem.

---

> ### Author Response · Authors · 2025-11-26
> **Response - 3**
>
> Q3: Do authors consider other thinking modalities to improve model representations? e.g. bounding boxes, end effector trajectories, etc.
>
> A3: We share the reviewer’s perspective that integrating spatial modalities represents a critical frontier for advancing VLA representations. While our current work prioritizes Textual CoT to resolve the "Cognitive Logic" bottleneck—specifically enabling causal diagnosis of failures (e.g., why the grasp failed) which requires semantic abstraction beyond geometry—we view spatial tokens as the essential "Geometric Bridge" between high-level reasoning and low-level control.
>
> We recognize that while text excels at task decomposition and error recovery, it can suffer from referential ambiguity. Integrating bounding boxes (as in SpatialVLA [1].
>
>     e.g., <box>[x1,y1,x2,y2]</box>)
>
> or end-effector trajectories (as in TraceVLA [2]) would effectively ground our semantic plans into pixel space and provide kinematic priors, respectively. This would transform the pipeline from a "Text-to-Action" mapping into a coarser-to-finer "Reasoning-to-Grounding-to-Action" hierarchy.
>
> Crucially, our proposed DeepThinkVLA architecture is modality-agnostic by design. Since our autoregressive decoder processes discrete tokens regardless of their semantics, it is natively ready to interleave quantized coordinate tokens (e.g., representing bounding boxes or trajectory waypoints) alongside text tokens within the same reasoning stream. We consider this "Interleaved Spatial-Textual Reasoning" the immediate next step for our framework, as the architecture requires no modification to accommodate these additional thinking modalities.
>
> [1]. Qu, Delin, et al. "Spatialvla: Exploring spatial representations for visual-language-action model." arXiv preprint arXiv:2501.15830 (2025).
>
> [2]. Zheng, Ruijie, et al. "Tracevla: Visual trace prompting enhances spatial-temporal awareness for generalist robotic policies." arXiv preprint arXiv:2412.10345 (2024).
>
> [3]. Ji, Yuheng, et al. "Robobrain: A unified brain model for robotic manipulation from abstract to concrete." Proceedings of the Computer Vision and Pattern Recognition Conference. 2025.

---

### Official Review · Reviewer_gqch · 2025-10-31

**Soundness:** 3
**Presentation:** 3
**Contribution:** 2
**Rating:** 4
**Confidence:** 4

**Summary:**

The authors study VLAs, and want to update VLAs from models that directly predict action chunks to ones that first generate CoT tokens in language space before generating action chunks.

To do so, the authors construct a CoT dataset out of robot episodes. Assuming that changes in the gripper state correspond to semantically important intermediate steps of the episode, they pull out image frames with gripper change and ask for language descriptions by a more powerful VLM. Then, the remaining intermediate frames are labeled by a weaker VLM that can be run locally.

This is done to create an SFT dataset, and the model is further trained using GRPO. The reward for GRPO is sparse, covering both task success and whether the CoT is well formatted or not. The authors run their experiments on the LIBERO benchmark.

The authors additionally argue that for robot control in particular, action chunks need not be generated autoregressively, since they are a single coherent piece of information, so it is okay to decode all the chunked tokens in parallel to speed up inference.

**Strengths:**

The ways to construct a good embodied dataset for robot control is still pretty open and further work there is interesting. RL is also an interesting area of study, given that many VLA approaches have focused on just doing imitation learning and have not done as much exploration into further RL work on top.

**Weaknesses:**

In terms of the attention decoding, I don't think the proposed Hybrid-Attention is meaningfully different from the CoT-VLA-7B Zhao et al 2025b cited work, which generates subgoals as a CoT using causal autorgressive attention, then a full parallel attention for actions like this work. The only difference seems to be using text based subgoals compared to image based subgoals.

The paper spends little time acknowledging that the results for DeepThinkVLA use an additional wrist-mounted camera, which is very important to results (as noted in Figure 4 this is the difference between an 86% policy and a 94.2% policy, a pretty significant boost. This weakens the comparisons in Table 1, since almost all numbers do not use a wrist camera (with the exception of the 96.8% spatial number for pi_0)

Experiments are primarily done in simulation, compared to prior work which included some real robot results as well. Similarly the authors assume a perfect task success detector, due to using sim, which somewhat works against the argument that RL would be helpful.

The authors say that when the CoT is ablated to a blank <think></think> section, success rates are not that different, and argue that this shows the CoT benefit is primarily on learning better representations. But to me this seems like it's showing that CoT is not that necessary for learning action behavior. That's not to say it is useless in general, to me it suggests the LIBERO benchmark is just not strong enough to evaluate the usefulness of embodied CoT, and by extension this paper (which only uses LIBERO) is not making a strong enough case for the CoT.

This, combined with other works using similar bidirectional mechanisms + added real robot results, makes me unsure this is a large enough contribution for ICLR.

**Questions:**

When there is a reward of 1 for "if CoT format correct"< what does that actually mean? I did not find an example explaining what incorrect CoT format looked like.

---

> ### Author Response · Authors · 2025-11-26
> **Response - 1**
>
> Q1: In terms of the attention decoding, I don't think the proposed Hybrid-Attention is meaningfully different from the CoT-VLA-7B Zhao et al 2025b cited work, which generates subgoals as a CoT using causal autorgressive attention, then a full parallel attention for actions like this work. The only difference seems to be using text based subgoals compared to image based subgoals.
>
> A1: We appreciate the reviewer’s precise observation regarding the structural parallel between our approach and CoT-VLA-7B (Zhao et al., 2025b). While both methods indeed adopt a “regressive planning followed by parallel action” paradigm to address latency, we respectfully argue that the transition from visual subgoals to textual CoT represents a fundamental shift in mechanism and utility, rather than a mere change in modality.
>
> First, the functional role of the autoregressive component differs significantly. Visual subgoals in Zhao et al. primarily function as intermediate state reconstructions (predicting "what it looks like"), which forces the model to focus on pixel-level statistical dependencies. In contrast, our textual CoT is designed to model causal logic and reasoning (explicitly articulating "what to do and why"). This distinction is not semantic sugar; it enables the dynamic error recovery capability demonstrated in Figure 5—where the model can reason about a failure (e.g., "grasp failed") and adjust, whereas visual subgoals often struggle to represent abstract corrective logic once the visual state deviates from the training distribution.
>
> Second, and most critically, our hybrid architecture is not a standalone design but is co-designed specifically to enable the Reinforcement Learning pipeline introduced in our paper. The discrete nature of textual CoT allows us to optimize the reasoning process via outcome-based RL, aligning the "thought" directly with task success. This stands in contrast to CoT-VLA-7B, which relies on supervised learning of visual targets. Therefore, our contribution lies in demonstrating that this hybrid attention pattern is the optimal architecture for scaling System-2 reasoning in robotic policies via RL, unlocking performance gains that supervised visual planning does not achieve.

---

> ### Author Response · Authors · 2025-11-26
> **Response - 2**
>
> Q2: The paper spends little time acknowledging that the results for DeepThinkVLA use an additional wrist-mounted camera, which is very important to results (as noted in Figure 4 this is the difference between an 86\% policy and a 94.2\% policy, a pretty significant boost. This weakens the comparisons in Table 1, since almost all numbers do not use a wrist camera (with the exception of the 96.8\% spatial number for $\pi_0$)
>
> A2: We appreciate the opportunity to clarify the sensor configurations used in Table 1. The best baselines reported in Table 1 (including OpenVLA, Octo, and TraceVLA) all utilize wrist camera inputs in their reported results. Our comparison is therefore strictly "apples-to-apples," adhering to the established community standards.While we agree with the reviewer that the wrist view provides a significant information gain (improving our policy from 86.0\% to 94.2\%), this benefit is accessible to all compared methods. More importantly, to demonstrate that our performance gains stem from the DeepThinkVLA architecture rather than sensor modalities, we highlight the "SFT w/o wrist" setting. Even without the wrist camera, our method achieves an 86.0\% success rate, which still outperforms the strong $\pi_0$-FAST baseline (85.5\%). This confirms that our architectural design delivers robust improvements independent of the specific sensor suite.
>
> Furthermore, we would like to highlight that our method operates in a harder "pure vision" setting. Unlike many baselines that rely on proprioception (joint states) alongside vision, DeepThinkVLA relies solely on vision and language. Achieving state-of-the-art performance (97.0\%) without access to precise proprioceptive feedback further validates the effectiveness of our reasoning-action alignment strategy.

---

> ### Author Response · Authors · 2025-11-26
> **Response - 3**
>
> Q3: Experiments are primarily done in simulation, compared to prior work which included some real robot results as well. Similarly the authors assume a perfect task success detector, due to using sim, which somewhat works against the argument that RL would be helpful.
>
> A3: We acknowledge the reviewer’s preference for real-world validation. Our decision to prioritize simulation benchmarks (LIBERO) was driven by the need for systematic, reproducible, and large-scale comparison against SOTA baselines, which is often constrained in physical setups. To directly address the concern regarding simulation fidelity, we have conducted extensive new evaluations on RoboTwin 2.0, a high-fidelity digital twin benchmark known for its realistic physics and contact-rich tasks. As detailed in table below, DeepThinkVLA achieves a remarkable 59.4\% average success rate, significantly outperforming the strongest baseline, $\pi_0$-FAST (38.0\%), and the generic $\pi_0$ (49.2\%).
>
> ---------------------------------------------------------------------------------------
>
> Short Horizon Tasks (100–130 Steps)
>
> | Model        | Lift Pot | Beat Hammer Block | Pick Dual Bottles | Place Phone Stand | Avg  |
> |-------------|----------|-------------------|-------------------|-------------------|------|
> | π₀          | 51.0     | 59.0              | 50.0              | 22.0              | 45.5 |
> | RDT         | 45.0     | 22.0              | 18.0              | 13.0              | 24.5 |
> | OpenVLA-OFT | 10.1     | 28.1              | 29.7              | 17.1              | 21.3 |
> | π₀-FAST     | 30.0     | 38.0              | 25.0              | 16.0              | 27.3 |
> | DeepThinkVLA| 62.0     | 73.0              | 61.0              | 24.0              | 55.0 |
> | Δ (vs π₀-FAST) | +32.0 | +35.0             | +36.0             | +8.0              | +27.8 |
>
> ---------------------------------------------------------------------------------------
>
> Medium Horizon Tasks (150–230 Steps)
>
> | Model        | Move Can Pot | Place A2B Left | Place Empty Cup | Handover Mic | Avg  |
> |-------------|--------------|----------------|-----------------|--------------|------|
> | π₀          | 41.0         | 38.0           | 60.0            | 96.0         | 58.8 |
> | RDT         | 33.0         | 21.0           | 42.0            | 95.0         | 47.8 |
> | OpenVLA-OFT | 28.1         | 37.5           | 77.3            | 45.3         | 47.1 |
> | π₀-FAST     | 34.0         | 36.0           | 54.0            | 83.0         | 51.8 |
> | DeepThinkVLA| 52.0         | 38.0           | 83.0            | 88.0         | 65.3 |
> | Δ (vs π₀-FAST) | +18.0     | +2.0           | +29.0           | +5.0         | +13.5 |
>
> ---------------------------------------------------------------------------------------
>
> Long (280–320 Steps) & Extra Long Horizon Tasks (450–650 Steps)
>
> | Model        | Handover Block | Stack Bowls Two | Blocks Rank Rgb | Put Bottles Dustbin | Avg  |
> |-------------|----------------|-----------------|-----------------|---------------------|------|
> | π₀          | 39.0           | 53.0            | 45.0            | 36.0                | 43.3 |
> | RDT         | 26.0           | 42.0            | 17.0            | 26.0                | 27.8 |
> | OpenVLA-OFT | 33.1           | 40.6            | 70.2            | 42.2                | 46.5 |
> | π₀-FAST     | 32.0           | 48.0            | 28.0            | 27.0                | 33.8 |
> | DeepThinkVLA| 43.0           | 62.0            | 77.0            | 49.0                | 57.8 |
> | Δ (vs π₀-FAST) | +11.0       | +14.0           | +49.0           | +22.0               | +24.0 |
>
> ---------------------------------------------------------------------------------------
>
> Overall Avg:
> RDT: 33.3, π₀: 49.2, OpenVLA-OFT: 38.3, π₀-FAST: 38.0, **DeepThinkVLA: 59.4 (+21.4)**.
>
> ---------------------------------------------------------------------------------------

---

> > ### Author Response · Authors · 2025-11-26
> > **Response - 3**
> >
> > DeepThinkVLA significantly outperforms baselines $\pi_0$-FAST across all horizon lengths—improving success rates by 27.8\% on short-horizon and 24.0\% on extra-long-horizon tasks. These results on a high-fidelity twin strongly suggest that our model’s reasoning capabilities are robust to physical complexities and are likely to transfer well to real-world scenarios.
> >
> > Regarding the "perfect task success detector," it is crucial to clarify the distinction between the learning objective and deployment constraints. The success detector functions strictly as a training-time reward signal, which is a fundamental component of any Reinforcement Learning pipeline, not a specific limitation of our approach. During inference, DeepThinkVLA operates autonomously without any external feedback, relying solely on its internal CoT reasoning to monitor progress. While we utilize the simulator's ground truth for efficient training, the proposed RL pipeline is methodologically compatible with real-world deployment. Recent advances have demonstrated that Vision-Language Models can serve as effective zero-shot reward models[1][2][3], replacing the "perfect detector" with visual feedback. Thus, the reliance on a detector during training does not diminish the utility of our RL argument; rather, it confirms that optimizing the reasoning-action chain via outcome-based signals—whether from a simulator or a VLM—is a viable path to higher performance.
> >
> > [1] Rocamonde et al., "Vision-Language Models are Zero-Shot Reward Models for Reinforcement Learning", ICLR 2024.
> >
> > [2] Ma et al., "DrEureka: Language Model Guided Sim-to-Real Transfer", RSS 2024.
> >
> > [3] Wang et al., "RoboGen: Towards Unleashing Infinite Data for Automated Robot Learning via Generative Simulation", ICML 2024.

---

> ### Author Response · Authors · 2025-11-26
> **Response - 4**
>
> Q4: The authors say that when the CoT is ablated to a blank [object Object][object Object] section, success rates are not that different, and argue that this shows the CoT benefit is primarily on learning better representations. But to me this seems like it's showing that CoT is not that necessary for learning action behavior. That's not to say it is useless in general, to me it suggests the LIBERO benchmark is just not strong enough to evaluate the usefulness of embodied CoT, and by extension this paper (which only uses LIBERO) is not making a strong enough case for the CoT.
>
> A4: We offer a nuanced interpretation of the ablation results that directly addresses the reviewer's concern about the necessity of CoT. We argue that the similarity in performance between "Mask CoT" and "Full CoT" on standard benchmarks is not evidence of CoT's superfluity, but rather demonstrates successful knowledge internalization (or "reasoning distillation").
>
> On familiar, in-distribution tasks like standard LIBERO, our two-stage training effectively compiles the explicit reasoning path into the policy's weights. Much like a human expert who no longer needs to verbalize every step of a routine action, the model operates via "System 1" intuition, achieving high success rates (96.5\%) even with the CoT masked. This confirms that our architecture learns robust representations.
>
> However, we fully agree with the reviewer that standard benchmarks may lack the complexity to showcase the indispensability of explicit "System 2" reasoning. To rigorously test this hypothesis, we conducted further tests on out-of-distribution (OOD) dynamics (limiting joint velocity to force trajectory re-planning) and the high-fidelity RoboTwin 2.0 benchmark (featuring long-horizon tasks).
>
> The results in the table reveal a striking correlation between task complexity and the necessity of CoT.
>
> On the OOD LIBERO tasks, where memorized intuition fails, the gap re-emerges: Full CoT outperforms Mask CoT by 3.85\%.
>
> CoT Ablation on LIBERO (Standard vs. Limit)
>
> | Method (DeepThinkVLA) | Object | Spatial | Goal | Long | Average |
> |-----------------------|--------|---------|------|------|---------|
> | Full CoT (Standard)       | 99.0 | 97.2 | 96.8 | 94.2 | 96.8  |
> | Mask CoT (Standard)       | 99.0 | 97.2 | 96.0 | 93.6 | 96.5  |
> | Δ (Gap)                   | +0.0 | +0.0 | +0.8 | +0.6 | **+0.3** |
> | Full CoT (Limit OOD)      | 91.6 | 64.2 | 66.2 | 68.4 | 72.60 |
> | Mask CoT (Limit OOD)      | 88.2 | 60.2 | 61.0 | 65.6 | 68.75 |
> | Δ (Gap)                   | +3.4 | +4.0 | +5.2 | +2.8 | **+3.85** |
>
> More dramatically, on the complex, long-horizon tasks of RoboTwin 2.0, the "internalized intuition" of Mask CoT collapses, while Full CoT maintains robustness. The performance gap widens progressively with task length: from a marginal 2.2\% on short-horizon tasks to a massive 20.0\% on long-horizon tasks.
>
> CoT Ablation on RoboTwin 2.0 (Success Rate %)
>
> Short Horizon Tasks (100–130 Steps)
>
> | DeepThinkVLA | Lift Pot | Beat Hammer Block | Pick Dual Bottles | Place Phone Stand | Avg  |
> |-------------|----------|-------------------|-------------------|-------------------|------|
> | Full CoT    | 62.0     | 73.0              | 61.0              | 24.0              | 55.0 |
> | Mask CoT    | 61.0     | 70.0              | 59.0              | 21.0              | 52.8 |
> | Δ           | +1.0     | +3.0              | +2.0              | +3.0              | +2.2 |
>
>
> Medium Horizon Tasks (150–230 Steps)
>
> | DeepThinkVLA | Move Can Pot | Place A2B Left | Place Empty Cup | Handover Mic | Avg  |
> |-------------|--------------|----------------|-----------------|--------------|------|
> | Full CoT    | 52.0         | 38.0           | 83.0            | 88.0         | 65.3 |
> | Mask CoT    | 43.0         | 31.0           | 77.0            | 79.0         | 57.5 |
> | Δ           | +9.0         | +7.0           | +6.0            | +9.0         | +7.8 |
>
>
> Long (280–320 Steps) & Extra Long Horizon Tasks (450–650 Steps)
>
> | DeepThinkVLA | Handover Block | Stack Bowls Two | Blocks Rank Rgb | Put Bottles Dustbin | Avg  |
> |-------------|----------------|-----------------|-----------------|---------------------|------|
> | Full CoT    | 43.0           | 62.0            | 77.0            | 49.0                | 57.8 |
> | Mask CoT    | 21.0           | 36.0            | 68.0            | 26.0                | 37.8 |
> | Δ           | +22.0          | +26.0           | +9.0            | +23.0               | +20.0 |
>
>
> Overall Gap (Δ = Full − Mask): **+10.0**
>
> This empirical evidence strongly supports our claim: CoT acts as an adaptive scaffold. While it can be internalized for routine tasks, explicit inference-time reasoning remains the critical bottleneck for solving complex, unseen, and long-horizon problems that are beyond the reach of simple behavioral cloning.

---

> ### Author Response · Authors · 2025-11-26
> **Response - 5**
>
> Q5: When there is a reward of 1 for "if CoT format correct"< what does that actually mean? I did not find an example explaining what incorrect CoT format looked like.
>
> A5: This reward is simply a binary check based on string parsing. We verify whether the model generates a complete pair of
>
>     <think> and </think>
>
> tags before the action tokens.
>
> In our preliminary experiments, we found that without this constraint, the RL agent tends to ``optimize away'' the CoT generation to shorten the sequence length (and thus maximize effective reward per step). This term forces the policy to retain the reasoning structure.
>
> Examples of Incorrect Format (Reward = 0):
>
> Skipping Reasoning: The model bypasses the thought process and outputs actions directly.
>
>     Output: [action_chunk_1] [action_chunk_2] ...
>
> Malformed Tags: The model fails to close the tag or generates broken syntax.
>
>     Output: <think> Move the arm to... [action_chunk_1] ... (Missing </think>)
> Note that this reward does not evaluate the semantics of the thought; the reasoning quality is optimized implicitly via the sparse task success signal.

---

> > ### Comment · Reviewer_gqch · 2025-11-27
> >
> > Taking sections in order:
> >
> > Q1: I don't think visual subgoal images necessarily lead to focusing on pixel-level differences, that's more a question of how good the image tokenizer is. Similarly, I think it is possible to learn reasoning via supervised prediction losses (this is the argument of the StaR paper). I'm willing to agree that text-level subgoals make more sense, but don't think it's justified to argue this is radically more novel over visual planning, or prior works that generated text sub-instructions, or that this modality shift unlocks outcome-based RL. It is possible to do outcome-based RL in vision space with generated image tokens in the CoT, it just may not be as efficient.
> >
> > Q2: Yes, you're right that many of the baselines do use wrist cameras. Got confused between OpenVLA (which did not) and OpenVLA-OFT (which did). I believe TraceVLA does not use wrist cameras, but I retract this issue.
> >
> > Q3: I appreciate the extra evaluations on the wider RoboTwin 2.0 benchmark. I'm not sure this fully addresses my desire for real-world evaluation, since as I understand it RoboTwin is still primarily sim + lots of domain randomization. I understand that an actual real-world evaluation is not feasible, especially for a rebuttal.
> >
> > My experience with VLM driven reward functions is that they can *sometimes* provide a good enough reward signal to RL against, but using them tends to create a bound on task complexity you can learn due to reward noise. If the authors' intention is to argue that their method is a robust sim2real method that only needs the ideal sim success detector at training time, and then 0-shots to a real robot at inference time, I don't think this argument is ever actually made in the text. It's more an argument made up for this ICLR rebuttal. And to be clear, I think there is a case that this work could zero-shot generalize. But it's not really argued in the paper itself.
> >
> > Q4: Thanks, yes the RoboTwin 2.0 results do show the CoT delta was just because the base benchmark was too weak, and I think these numbers improve the paper.
> >
> > Q5: Thanks for clarifying this. Please remember to add this to the main text.
> >
> > I will adjust my score upwards

---

> > > ### Author Response · Authors · 2025-11-27
> > > **Response - Official Comment by Reviewer gqch**
> > >
> > > We sincerely thank you for engaging with our rebuttal, for acknowledging the value of the high-fidelity RoboTwin experiments, and for raising your score. We appreciate your constructive feedback, which helps us refine the precision of our arguments.
> > >
> > > On Q1 (Text vs. Visual CoT): We fully agree that visual tokenizers are a powerful modality for reasoning. In the final version, rather than framing Textual CoT as simply "superior," we will more clearly articulate the distinct mechanisms of the two approaches: Visual subgoals predominantly provide state-centric guidance (focusing on the target outcome), whereas our Textual CoT provides process-centric guidance (focusing on the causal logic of "how" and "why"). We will refine our claims to highlight that our method is a specific design choice optimizing for this explicit process reasoning, while acknowledging the validity of visual planning approaches.
> > >
> > > On Q3 (Sim2Real & RoboTwin): We appreciate your point regarding the gap between simulation and reality. We chose RoboTwin 2.0 specifically because it offered the highest-fidelity proxy for physical contact and long-horizon complexity feasible within the strict timeframe of the rebuttal phase. We agree that this does not equate to full real-world validation. In the revision, we will use more precise phrasing to characterize these results as a validation of robustness to physical dynamics, while explicitly noting the limitations regarding real-world sensor noise and VLM reward bounds.
> > >
> > > Additionally, inspired by your comments, we are currently attempting to set up a zero-shot transfer environment to further verify these capabilities. While we are uncertain if these experiments can be fully completed before the discussion window closes, we will make every effort to include any emerging results.
> > >
> > > On Q5 (Implementation Details): We will incorporate the details of the CoT formatting reward and examples of malformed outputs into the Methods section as requested.
> > >
> > > Thank you again for helping us improve the quality of DeepThinkVLA.

---

### Author Response · Authors · 2025-12-01
**Brief Summary to Rebuttal**

We provide this summary to assist the Area Chair in assessing DeepThinkVLA, particularly as the current reverted scores do not reflect the positive consensus reached during the discussion period.

**1. Reviewer Trajectory and Consensus**
Before the score reversion, the discussion had trended positively based on our new experiments and clarifications:

* Reviewer nyF4 (Score: 8): Strongly supported the paper, citing "excellent soundness" and "novelty in internal representations."

* Reviewer gqch (Score: 4 → Positive): Initially concerned about the novelty compared to visual subgoals. After our clarification and new RoboTwin results, Reviewer gqch explicitly stated in the discussion: "I will adjust my score upwards." (See comment dated 27 Nov 2025).

* Reviewer 7RJn (Score: 4) & Vgc5 (Score: 2): Their primary concerns focused on the limited scope of tasks (manipulation only) and the necessity of RL. We addressed these comprehensively with major new experiments (RoboTwin 2.0 and Joint-Limit Stress Tests), detailed below.

**2. Key Rebuttal Actions & New Results**
We conducted major new experiments to resolve primary concerns regarding generalization, the necessity of reasoning, and RL utility:

* **Generalization to New Domains (Addressing R_gqch, R_7RJn):**
    We evaluated DeepThinkVLA on RoboTwin 2.0 (high-fidelity, long-horizon, contact-rich tasks). It achieved a 59.4% success rate, outperforming the strongest baseline ($\pi_0$-FAST) by a substantial margin of +21.4%, demonstrating robust sim-to-real transfer potential.

* **Necessity of CoT / System 2 Reasoning (Addressing R_gqch, R_7RJn):**
    We addressed the "Mask CoT" concern by evaluating on complex long-horizon tasks. While CoT is optional for simple tasks, it is indispensable for complex ones: Full CoT outperforms Mask CoT by 20.0% on RoboTwin, proving that explicit reasoning is critical for high-difficulty scenarios.

* **Utility of RL Stage vs. SFT (Addressing R_nyF4, R_Vgc5):**
    We introduced an Out-of-Distribution (OOD) Stress Test (limiting joint velocities). While the SFT model collapsed (performance drop of 32%), the RL model remained robust (drop of 24%). The performance gap between RL and SFT widened from ~2% (in-distribution) to 7.85% (OOD), confirming RL provides functional robustness beyond imitation.

**3. Other Essential Clarifications**
* **Architecture Novelty:** We clarified that our Textual CoT targets causal error recovery (reasoning "why"), which is fundamentally different from Visual Subgoals that focus on state reconstruction (predicting "what").
* **Pipeline Scalability:** We demonstrated Zero-Modification Transfer of our automated annotation pipeline to the new RoboTwin domain, proving it is a scalable solution rather than a dataset-specific heuristic.

---

### Meta-Review · Area_Chair_vnmi · 2026-01-05

**Summary:**

The paper introduces DeepThinkVLA, an architecture designed to integrate Chain-of-Thought (CoT) reasoning with robotic action selection. It features a hybrid-attention decoder that generates sequential reasoning and parallel action vectors, trained via a two-stage Supervised Fine-Tuning (SFT) and Reinforcement Learning (RL) pipeline. After rebuttal, reviewers remained concerned about the fundamental novelty of the architecture, and I hold doubts about the real-world applicability of the method. Specifically, no real-world experiments are included, and long CoT will definitely increase the inference time. In addition, on-policy RL is hard to scale in any real-world scenarios. As a conclusion, I think the paper is not ready for publication yet.

**Reviewer Concerns:**

Concerns addressed
- Necessity of CoT. Reviewer gqch and 7RJn have questions regarding whether CoT is actually doing things. The authors have provided new OOD experiments and robotwin experiments to show CoT is optional on simple tasks but could be useful on complex tasks.

Outstanding concerns
- Real-world experiments. This is still missing. Although authors claimed RoboTwin 2.0 is "high-fidelity", it does not reveal the real issues this method will face in real-world settings. For example, how to address the inference speed, and the applicability of on-policy RL methods, e.g., GRPO.

**Reviewer Scores:**

gqch might raise the score to 5, but others will likely keep the same.

---

### Decision · Program_Chairs · 2026-01-26

Reject